# ADAPTIVE CONTROL FOR TEST-TIME SCALING

## ABSTRACT

Large language models often rely on static computational budgets for reasoning, leading to suboptimal performance due to "underthinking" (premature termination) or "overthinking" (performance degradation from excessive computation). In this work, we demonstrate that models provide a rich, context-dependent signal about their internal cognitive state through the sub-argmax probabilities of special-purpose "control tokens.". We introduce a framework of **Adaptive Control Token Sampling (ACTS)** policies that leverage these probability spikes to dynamically regulate the generation process. Our experiments show that ACTS effectively mitigates underthinking on complex reasoning tasks. To avert the performance collapse caused by overthinking in naive policies, we propose an **Adaptive Self-Critique Sampler** that uses `EOT` spikes as triggers for self-evaluation, boosting reasoning accuracy upto $\sim 9.8\%$ on the MATH-500. On instruction-following tasks, ACTS leverages `EOS` spikes to improve the quality-efficiency trade-off. Finally, we used spikes to pioneer, a novel parallel sampling technique that intelligently spawns high-quality reasoning trajectories from a shared trace. Our work establishes control token probabilities as a powerful, untapped resource for creating more robust and efficient inference strategies, offering a low-cost method for test-time scaling.

## 1 INTRODUCTION

Modern Large Language Models (LLMs) tackle complex reasoning by generating explicit multi-step rationales, a technique known as Chain-of-Thought (CoT) prompting (Wei et al., 2022) that has become a cornerstone of state-of-the-art systems (Guo et al., 2025). The efficacy of this "slow-thinking" paradigm, however, is deeply intertwined with the length of the generated reasoning trace. While longer CoTs can provide necessary computational steps, they also introduce significant latency and are susceptible to error accumulation, a phenomenon often termed "overthinking" (Sui et al., 2025).

Recent foundational work has formalized this trade-off, demonstrating that reasoning performance does not scale monotonically with length but instead follows an inverted U-shaped curve (Wu et al., 2025). This establishes the existence of an *optimal CoT length* that is dependent on both task difficulty and model capability, challenging the naive assumption that more computation is always better. This insight has catalyzed a vibrant research area focused on *efficient reasoning*. Current solution paradigms largely fall into two categories. The first is **model-centric**, aiming to build innately more efficient reasoners through costly training-time interventions such as fine-tuning on optimal-length data (Yang et al., 2025b; Wu et al., 2025) or incorporating length-based penalties into reinforcement learning (Luo et al., 2025). The second paradigm is **inference-centric**, seeking to dynamically control existing models at test-time. A Seminal work in this area, S1 (Muennighoff et al., 2025), introduced budget forcing, a technique that behaviorally controls reasoning by either forcefully terminating or prolonging thinking with special tokens (e.g., "Wait"). Other inference-time methods often rely on auxiliary signals, such as scores from external reward models (Sun et al., 2024) or the consistency of latent embeddings (Wang et al., 2025).

While powerful, existing inference-time methods often depend on separately trained models or complex heuristics disconnected from the LLM's core generative process. In this work, we identify and leverage a more fundamental, previously under-explored signal for generation control: the sub-argmax probabilities assigned to control tokens like End-of-Sequence and End-of-Thinking tokens.

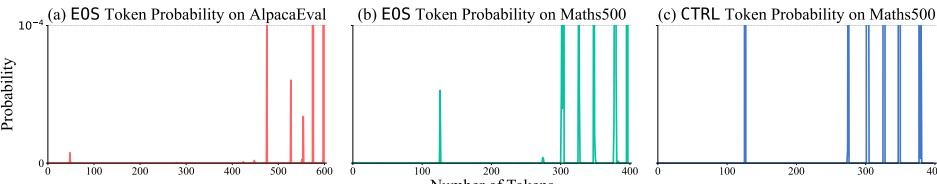

Figure 1: Token-wise probabilities of control tokens during LLM decoding, illustrating a duality in the signal. Subfigure (a) shows $P(t_{EOS})$ on AlpacaEval, where an initial phase of sparse spikes transitions to a denser, more regular pattern, suggesting a shift in the model's operational state towards termination. Subfigures (b) and (c) show post-thinking $P(t_{EOS})$ and in-thinking $P(t_{EOT})$ on MATH500, respectively, highlighting the sharp, transient nature of spikes that act as candidate completion points. The non-monotonic and state-dependent nature of these signals motivates adaptive control policies over static ones.

Our central empirical finding, illustrated in Figure 1, is that these internal termination signals are a rich, structured indicator of the model's readiness to conclude a generation phase.

To harness this novel signal, we introduce **A**daptive **C**ontrol for **T**est-time **S**caling (ACTS), a framework that casts adaptive generation control as a principled optimal stopping problem. This represents a shift from the behavioral control paradigm of Muennighoff et al. (2025) to a more fundamental, fine-grained probabilistic control. Our approach is notably simple and low-cost: it is **training-free** and can operate on a given LLM as long as token probabilities are accessible. Within this framework, an external control policy, $\pi_{ACTS}$, observes the sequence of control signal probabilities to determine the optimal moment to terminate. We derive a suite of stopping policies from distinct theoretical motivations, including evidence accumulation, predictive control, and a novel **adaptive self-critique policy** that uses the LLM to evaluate its own reasoning at critical junctures—an efficient, single-model realization of an actor-critic mechanism.

Our comprehensive experiments demonstrate that ACTS policies significantly outperform standard baselines. On the AlpacaEval benchmark, our methods reduce response length under budget constraints while maintaining high win-rates. On complex reasoning tasks like MATH500 and AIME, they achieve higher accuracy than greedy decoding by enabling more efficient and effective allocation of "thinking time." Our adaptive self-critique policy, in particular, establishes a new state of the art by intelligently resolving the dilemma of when to stop versus when to continue thinking.

## 1.1 OUR CONTRIBUTIONS

The primary contributions of this work are:

**1. We identify and characterize a novel class of control signals** within LLMs: the sub-argmax probability spikes of special tokens (e.g., EOS, EOT). We demonstrate that the dynamics of these signals reveal the model's internal state regarding termination and reasoning progress.

**2. We develop a framework of spike-aware control policies** that resolve the trade-off between reasoning depth and efficiency. Our policies mitigate underthinking by adaptively prolonging deliberation, while our novel Adaptive Self-Critique policy prevents performance degradation from overthinking by using spikes to trigger efficient self-evaluation.

**3. We demonstrate a superior trade-off between output quality and computational cost.** On instruction-following tasks, our policies reduce token count by up to 19% while improving response quality. On complex reasoning tasks, they achieve higher accuracy with fewer tokens than baselines.

**4. Parallel sampling from sequential chain-of-thought** We introduce a novel sampling method that uses a shared KV-cache to spawn multiple answer trajectories from a single reasoning trace. By using hesitation spikes and self-critique scores as principled triggers for forking, this approach achieves higher accuracy than strong sequential baselines while being significantly more computationally efficient than naive parallel sampling.

## 2 RELATED WORK

The challenge of enhancing LLM reasoning efficiency is an active area of research. A primary focus has been on mitigating the "overthinking phenomenon" (Sui et al., 2025), where models generate excessively long Chain-of-Thought (CoT) traces. Foundational work has established that reasoning performance follows a non-monotonic, inverted U-shaped curve with CoT length, motivating the search for an optimal, task- and model-dependent reasoning budget (Wu et al., 2025). Current approaches largely fall into two categories: **model-centric** methods that seek to build innately more efficient reasoners through training-time interventions like fine-tuning on optimal-length data (Yang et al., 2025b), and **inference-centric** methods that aim to dynamically control existing models at test-time. Our work, ACTS, belongs to the latter category.

Inference-time control strategies have evolved from behavioral interventions to signal-based termination. A seminal approach introduced "budget forcing," using special tokens like "Wait" to behaviorally prolong or terminate reasoning (Muennighoff et al., 2025). More recent methods have focused on early stopping in sampling-based decoding by leveraging auxiliary signals, such as scores from external reward models (Sun et al., 2024), the consistency of latent embeddings (Wang et al., 2025), or learned confidence scores (Huang et al., 2025). ACTS advances this paradigm by identifying and utilizing a more fundamental, natively available signal: the sub-argmax probability of control tokens ($t_{control}$). By framing the control task as a principled optimal stopping problem and designing policies grounded in established theoretical concepts—such as our novel, intra-model actor-critic mechanism for self-critique—our work provides a training-free, model-agnostic framework for more precise and efficient test-time control. A more comprehensive discussion of related literature is available in Appendix A.

## 3 PRELIMINARIES: THE ACTS FRAMEWORK

To address the control dilemma identified in our motivations, we introduce the ACTS framework. This section provides the formal groundwork for our approach. We first establish notation for the control signal, then we cast the problem of adaptive termination as one of principled policy design for a controlled stochastic process, drawing on concepts from optimal stopping theory.

We formally cast the problem of adaptive generation control as an optimal stopping problem for a controlled stochastic process. The objective is to design a **control policy**, $\pi_{ACTS}$, that observes the generative process and determines the optimal **stopping time**, $\tau$. The control policy $\pi_{ACTS}$ is not a passive observer but an active controller that intervenes in the generation process. Its actions guide the process until its chosen stopping time $\tau$ is reached.

**Continuing the Process** ($t < \tau$). If the policy's stopping rule has not been triggered, its action is to continue. This typically involves allowing the LLM to emit its native end-of-thinking token ($t_{EOT}$). If $\pi_\theta$ attempts to terminate prematurely by predicting $t_{EOT}$ as the argmax token, but $\pi_{ACTS}$ has not yet reached its stopping time, the controller intervenes by forcing the emission of a special "Wait" token, $t_{Wait}$. This control action, inspired by Muennighoff et al. (2025), keeps the system in a non-terminal state, allowing for further deliberation.

**Terminating the Process** ($t = \tau$). When the policy's stopping condition is met, the controller issues a terminal action. This involves forcing the emission of the appropriate control token, $t_{control}$, thereby halting the process.

This control-theoretic perspective clarifies that ACTS is a closed-loop system where the controller ($\pi_{ACTS}$) observes the system's output signal ($s_t$) and applies control actions ($t_{Wait}$ or $t_{control}$) to steer the system towards an optimal termination state. The specific methods we introduce next are different instantiations of this control policy, $\pi_{ACTS}$.

## 4 METHODOLOGY: A SPECTRUM OF PRINCIPLED GENERATION POLICIES

We propose a framework for dynamically steering the autoregressive generation of a Large Language Model (LLM) to mitigate common failure modes such as **underthinking** (premature convergence)

and **overthinking** (unproductive deliberation). Our approach, termed Adaptive Control of Token Sequences (ACTS), intervenes at each decoding step by monitoring the probability of specific *control tokens*. This control signal, $s_t = \pi_\theta(t_{\text{control}}|\mathbf{x}_{<t}, C)$, is interpreted by a suite of principled policies ($\pi_{\text{ACTS}}$) to solve the optimal stopping problem.

These policies range from simple, deterministic rules derived from principles of temporal integration and evidence accumulation, to adaptive strategies that leverage the LLM's own capacity for self-evaluation. This spectrum allows for a trade-off between computational simplicity and nuanced, context-aware control.

## 4.1 POLICIES FOR SAMPLING

We refer to an instance where the control signal exceeds a predefined threshold ($s_t > \delta$) as a **"spike."**

**N-Spike Counter Policy** This policy treats each spike as a discrete vote for termination. It requires a sustained signal, demanding the accumulation of $N_{\text{patience}}$ such votes before acting. This filters out single, potentially spurious, transient events and ensures the LLM's intent to terminate is stable over time. The policy (Algorithm 1) maintains a count of observed spikes. The stopping time $\tau$ is the first time step $t$ at which this count reaches a predefined evidence threshold $N_{\text{patience}}$.

**Adaptive Policies via Self-Critique** While robust, deterministic policies are context-agnostic. We introduce a more sophisticated class of policies inspired by **actor-critic methods**, where the LLM is leveraged as its own critic. The generative process is the **actor** ($\pi_\theta$), which produces reasoning, and the same LLM, prompted for self-evaluation, is the **critic**.

**Algorithm 1:**

*N-Spike Counter Policy*

1: **Input**: LLM $\pi_\theta$, Prompt $C$, $t_{\text{control}}$, Spike Probability Threshold ($\delta_{\text{count}}$), Spikes Count Threshold ($N_{\text{patience}}$)
2: Initialize spike_count $\leftarrow 0$; $\mathbf{x} \leftarrow []$.
3: **for** each generation step $t = 1, 2, \ldots$ **do**
4:    $s_t \leftarrow \pi_\theta(t_{\text{control}} \mid \mathbf{x}_{<t}, C)$   ▷ Control Token Probability
5:    **if** $s_t > \delta_{\text{count}}$ **then**
6:       spike_count $\leftarrow$ spike_count $+1$
7:    **end if**
8:    **if** spike_count $\geq N_{\text{patience}}$ **then**
9:       Emit $t_{\text{control}}$ and **return**   ▷ Sets stopping time $\tau \leftarrow t$
10:    **end if**
11:    Emit default token
12: **end for**

Instead of performing costly self-evaluation at every step, we use probability spikes as a trigger, identifying critical junctures where a critique is most valuable. The critic's output, a reasoning quality score $s = C(T_k) \in \{1, \ldots, 5\}$, then informs the generation decision.

**Parallel Trajectory Generation with follow-the-leader Consensus** We note that with the same reasoning trace, we can fork answer generation at different points in the output process. Such forking ensures that the user may receive an early version of the answer which can be adapted as the thinking proceeds. In fact this is often how humans ourselves reason, having

we can fork the reasoning process at opportune moments to generate an ensemble of answers $\mathcal{A}$, with the final answer determined by majority vote. Self-critique ensures this computationally intensive strategy is used efficiently.

**Quality-Gated Forking.** A fork is initiated only when two conditions are met: (1) a hesitation spike is detected ($\max_{t_h \in T_{\text{hesitate}}} p(t_h|T_k) > \tau_p$) and (2) the self-critique score is high ($C(T_k) \geq s_{\text{fork}}$). This dual condition identifies states where the model is both uncertain about the next step and has produced a high-quality line of reasoning so far—an

**Algorithm 2:**

*Generic Adaptive Self-Critique Policy*

1: **Input**: LLM $\pi_\theta$, Prompt $C$, $t_{\text{control}}$, Spike Threshold ($\delta_{\text{critique}}$), Critique Prompt ($C_{\text{critique}}$)
2: **for** each generation step $t = 1, 2, \ldots$ **do**
3:    $s_t \leftarrow \pi_\theta(t_{\text{control}} \mid \mathbf{x}_{<t}, C)$   ▷ Control Token Probability
4:    **if** $s_t > \delta_{\text{critique}}$ **then**
5:       context $\leftarrow$ context $+C_{\text{critique}}$
6:       score $\leftarrow$ GenerateCritique(LLM, critique_context)
7:       **if** score $= 5$ **then**
8:          Emit $t_{EOT}$ and **return**  ▷ Set stopping time $\tau \leftarrow t$
9:       **else**
10:          Emit $t_{Wait}$   ▷ Critique failed, force continuation
11:       **end if**
12:    **else**
13:       Emit default token
14:    **end if**
15: **end for**

ideal point to explore alternative conclusions. The
primary branch continues generating, while parallel
branches are spawned to generate candidate answers
from the current state.

## 5 EXPERIMENTAL SETUP

**Datasets and Tasks**    We evaluate our method on a diverse suite of benchmarks targeting two key
capabilities: reasoning and instruction following. For **reasoning**, we use three benchmarks spanning
mathematical and logical problem-solving. First, for arithmetic reasoning, we use **GSM-8K** (Cobbe
et al., 2021), a collection of 1,320 grade-school math problems requiring multiple steps of basic
arithmetic. Second, we assess performance on more complex mathematical challenges using a
500-problem subset of the **MATH** benchmark (Hendrycks et al., 2021). Third, to test advanced
problem-solving, we include 30 competition-level questions from the **AIME 2025**. For **instruction
following**, we use **AlpacaEval** (Dubois et al., 2024), an automatic, LLM-based evaluation bench-
mark consisting of open-ended user queries from real-world scenarios.

**Models**    We conducted experiments across different model families and scales. For **reasoning
tasks**, we employ models from two distinct families. From the **Qwen3** series, recognized for
state-of-the-art performance on public leaderboards (Yang et al., 2025a), we select `Qwen3-4B`,
`Qwen3-8B`, and `Qwen3-14B`. Additionally, we use the `s1.1-7B` and `s1.1-32B` (Muennighoff
et al., 2025) models to broaden our evaluation. For the **instruction following task** on AlpacaEval,
we utilize the `Llama3.1-8B-Instruct` model.

**Evaluation Metrics**    Following standard practices for each task, we employ strict and established
metrics. For the reasoning benchmarks (GSM-8K, MATH, and AIME), we measure performance
using **Accuracy**, i.e. a model's prediction is correct only if it exactly matches the ground-truth
solution. For the instruction-following benchmark (AlpacaEval), we report the **win rate** and **length-
controlled win rate** against a strong reference model (e.g., GPT-4), as determined by an automated
GPT-4-based evaluator.

## 6 EXPERIMENTAL RESULTS

Our experiments are designed to address the following research questions:

**RQ1: Signal Characterization and Analysis:** What are the empirical properties of the sub-argmax
control token probability signal, $P(t_{control})$, across different models and task domains (Fig-
ure 1)?

**RQ2: Instruction Following Efficiency:** Can deterministic ACTS policies leverage the $P(t_{EOS})$
signal to reduce verbosity and improve the trade-off between response quality and computa-
tional cost on instruction-following tasks (e.g., on AlpacaEval under tight token budgets)?

**RQ3: Mitigating Reasoning Underthinking:** Can ACTS policies, intervene on premature end-of-
thinking spikes, to improve the reasoning accuracy of models?

**RQ4: Analysis of Overthinking:** Does a naive or overly conservative policy of prolonging rea-
soning lead to diminishing returns or performance degradation, thereby confirming the over-
thinking problem?

**RQ5: A new parallel sampling technique:** Can a shared sequential reasoning trace be used to
spawn multiple parallel traces for generation? We address this question in this subsection.

### 6.1 RQ1: SIGNAL CHARECTARIZATION

We study three distinct type of control tokens: **end-of-sentence spikes**, **end-of-thinking spikes**, and
**hesitation spikes**. These signals correspond to concluding a response, finalizing a complex thought,
or actively reasoning through a problem.

**End-of-Sentence Spikes for Early Termination**    We observe that the model emits distinct spikes
for the End-of-Text (`EOS`) token at semantically appropriate completion points, as illustrated in

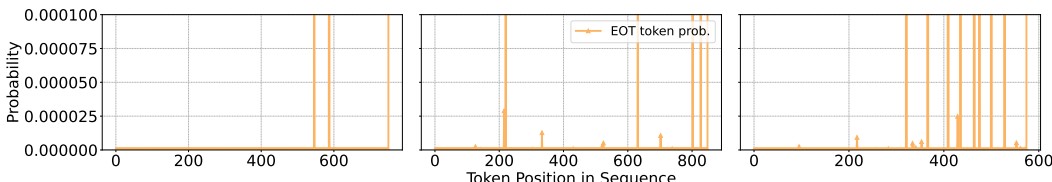

Figure 2: End-of-Sentence (EOS) Spikes for `Llama3.1-8b-Instruct` for AlpacaEval2.0

Figure 2. While the probability remains near zero for most of the generation, it exhibits sharp, low-magnitude spikes (up to ∼0.0001) that often align with the natural conclusion of a sentence or a complete answer. These spikes serve as a soft signal that the model has fulfilled the user's request. These serve as appropriate points to implement early termination strategies.

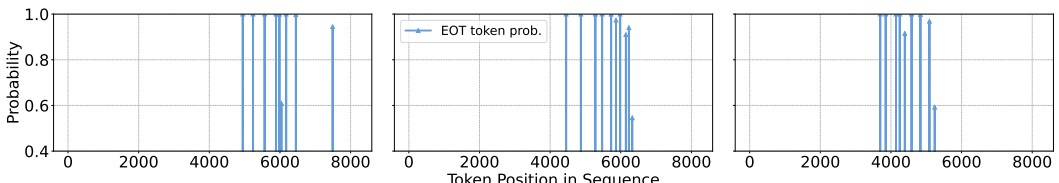

Figure 3: End-of-Thinking (EOT) Spikes for `Qwen3-8B` for Maths-500

**End-of-Thinking Spikes for Elongating Reasoning** For complex reasoning tasks, a common failure mode is *underthinking*, where the model provides a premature or superficial answer. These spikes, generated during a reasoning task (see Fig. 3), displays high-probability (approaching 1.0) spikes for the EOT token. Unlike the subtle end-of-sentence signals, these are periodic indicators that the model considers its chain of thought complete. By configuring the generation process to continue until such a spike is detected, we can encourage the model to "think longer" and develop a more thorough solution, helping to prevent *underthinking*.

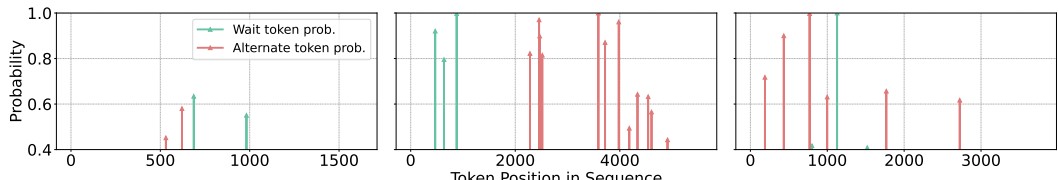

Figure 4: Hesitation Spikes for `Qwen3-8B` for Maths-500

**Hesitation Spikes for Reasoning Path Correction** During problem-solving, it is beneficial for the model to pause, reconsider, and explore different approaches. We analyze two special tokens, `Wait` and `Alternatively`, designed to facilitate this behavior. Figure 4 shows that the model utilizes `Wait` (green) and `Alternate` (red) tokens with high probability during complex reasoning. A dense cluster of `Alternate` spikes, for instance, suggests a moment of significant reconsideration. These hesitation tokens are not signs of failure but rather functional components of a robust reasoning process, enabling the model to self-correct and navigate complex problem spaces.

## 6.2 RQ2: INSTRUCTION FOLLOWING EFFICIENCY:

Verbosity is a common phenomenon in LLM generation. Simple answers may be elongated due to reward hacking on length. But can these effects be mitigated using end-of-sequence spike control? We tackle this query by evaluating our samplers on the AlpacaEval 2.0 benchmark, measuring

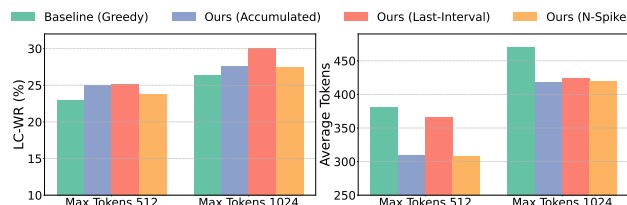

Figure 5: Comparison of Response Quality (LC-WR) and Generation Cost (Avg. Tokens) of our spike-aware samplers vs. baseline on AlpacaEval.

both quality via Length-Controlled Win Rate (LC-WR) and cost via average generated tokens.

The results, presented in Figure 5, reveal a clear Pareto improvement over the greedy decoding baseline. Our **Last-Interval Budget Sampler**, which strategically terminates generation based on prominent `[EOS]` spikes, achieves an LC-WR of 30.11%, a significant +3.7% gain over greedy decoding. This quality enhancement is coupled with an efficiency gain of 11–19%.

### 6.3 RQ3: MITIGATING REASONING UNDERTHINKING

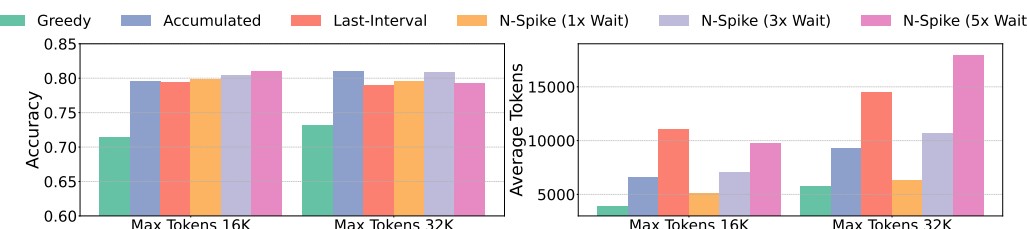

Figure 6: Analysis of spike-aware stopping policies on the MATH-500 benchmark. These policies use probability spikes of the `[EOT]` token to determine when to halt reasoning. (a) All spike-aware policies overcome the underthinking of the Greedy baseline, improving accuracy. (b) The choice of policy dictates a trade-off between token cost and the risk of overthinking.

Under-thinking on complex tasks is known to suppress the performance of reasoning models on hard benchmarks. But can this be modulated through control tokens? Specifically, underthinking occurs when a model terminates its reasoning chain prematurely, before reaching a correct solution that was possible for the model to reach. We investigate our samplers that effectively combat under-thinking, on the MATH-500 benchmark, with results in Figure 6.

While the greedy baseline succumbs to under-thinking, achieving the lowst accuracy on the benchmark, our control-token aware samplers deliver significant performance gains, simply by leveraging the information available in the end-of-thinking spike probabilities. While Accumulated sampler waits for the total of the end-of-thinking token probabilities to reach a threshold, the n-spike sampler waits for 'n' spikes of at least a certain probability threshold. We find that the simple 5-spike sampler which waits for 5 probability spikes before exiting the chain-of-thought performans the best on this benchmark. The more complex last-interval sampler underperforms both on token-budget as well as accuracy versus the simpler alternatives.

**Note on overthinking:** While the 5-spike sampler achieves great results in the 16K thinking limit setting, we notice that the accuracy drops from 81% to 79.2% on a larger 32K context budget This indicates that simply extending the thinking trace may sometimes lead to poorer results.

## 7 ANALYSIS OF OVERTHINKING: AVERTING OVERTHINKING IN COMPLEX REASONING

Overthinking may not just lead to greater cost in terms of compute budgets and wasted tokens, but also lead to lower downstream performance. Hence a naive policy simply suppressing the end-of-thinking tokens does not work. This leads to the question: is there an adaptive method that overcomes the weaknesses of these naive deterministic samplers, that can mitigate overthinking?

Our experiments, summarized in Table 1, address this very question. We note that the best-performing sampler is the adaptive self-critique sampler. Specifically, at each spike, the model is forced to answer the query: "Is this reasoning trace correct, answer on the scale of 1-5?" wherein it responds with a single token, that then determines whether the answer continues or not.

Specifically, on the S1-32B model, the self critique is invoked as follows. When there is a spike on the end-of-thinking token, the sampler initiates a critique to determine if the current reasoning is sound or if more work is needed. This intelligent, state-dependent approach achieves superior performance over the naive samplers. While the performance gets boosted by up to 3% over the best-performing deterministic baseline, we also see gains of up to 50% on reasoning tokens over the

Table 1: Ablation on handling `[EOT]` spikes. A naive, fixed policy of suppressing the spike leads to overthinking and performance collapse (e.g., S1 models at 5x Wait). Our Self-Critique Sampler uses `[EOT]` spikes as a trigger for adaptive evaluation, achieving superior accuracy and efficiency.

| Dataset | Model | Sampler (EOT Handling Policy) | Accuracy ↑ | Avg. Token Count ↓ |
|---|---|---|---|---|
| MATH-500 | S1 (32K) | Baseline | 0.732 | 5752.40 |
| | | Naive (Wait 1x) | 0.796 | 6317.92 |
| | | Naive (Wait 3x) | 0.808 | 10659.59 |
| | | Naive (Wait 5x) | 0.792 | 17906.58 |
| | | **Self-Critique (Adaptive)** | **0.822** | **8885.29** |
| AIME 2025 | S1-32B | Baseline | 0.400 | 9552.75 |
| | | Naive (Wait 1x) | 0.466 | 10653.63 |
| | | Naive (Wait 3x) | 0.567 | 14503.36 |
| | | Naive (Wait 5x) | 0.537 | 17786.21 |
| | | **Self-Critique (Adaptive)** | **0.567** | **12345.67** |
| | Qwen3-8B | Baseline | 0.700 | 18154.00 |
| | | Naive (Wait 3x) | 0.700 | 19007.30 |
| | | Naive (Wait 5x) | 0.700 | 20292.45 |
| | | Naive (Wait 7x) | 0.737 | 20398.96 |
| | | **Self-Critique (Adaptive)** | **0.767** | **19269.56** |

best deterministic baseline. This shows the need to not just use control token probabilities during generation, but also, appropriately at spikes, invoke the models own self-critique capabilities to decide whether to continue thinking or not.

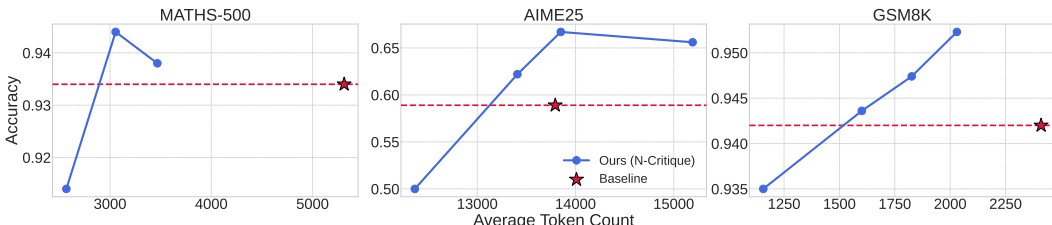

Figure 7: Accuracy vs. Computational Cost Trade-off for Qwen-3 8B. This figure illustrates the performance of our N-Critique sampler method compared to the baseline. Each blue point on the curve represents a policy that halts after N critiques (N=1, 3, 5, or 7). (a) Performance on Maths-500. (b) AIME25 (c) GSM8K

**Adaptive self-critique to guide rethinking** While the adaptive self-critique may be invoked during end-of-thinking spikes, it may also be invoked during other rethinking token spikes like "Wait" or "Alternatively". These signals can then be used to trigger a self-critique module which evaluates the reasoning chain and provides a confidence score. Generation is terminated only when the model expresses high confidence in its current path.

To check for different confidence levels on such a self-critique we can require the model to get a particular critique score n times before termination, a method we term `N-Critique Sampler (N)`. This prevents premature termination based on a single, potentially spurious confidence spike. We evaluate this approach on the MATH-500, AIME25, and GSM8K benchmarks using Qwen-3 8B and 14B models. The results are detailed in Figure 7 for Qwen3 8-B and in Figure 8 for Qwen3 14-B.

On the MATH-500 dataset, for both the 8B and 14B models, `N-Critique Sampler (3)` surpasses the baseline accuracy (94.4% vs. 93.4% for 8B; 95.0% vs. 94.0% for 14B) while simultaneously reducing the average token count by over 35%. On the more challenging AIME25 benchmark, which requires more extensive reasoning, the performance scales with $N$, with `N-Critique Sampler (7)` achieving a top accuracy of 72.2%, an improvement over the 65.5% baseline. The 8B model shows a similar trend, peaking at $N = 5$ with 66.7% accuracy compared to the 58.9% baseline. Finally, on GSM8K, the benefits are again pronounced. The self-critique policies consistently outperform the baseline in accuracy while drastically reducing token usage. For the 14B model, `N-Critique Sampler (7)` achieves an outstanding 95.5% accuracy using only 1743 tokens, compared to the baseline's 94.2% accuracy at 1908 tokens.

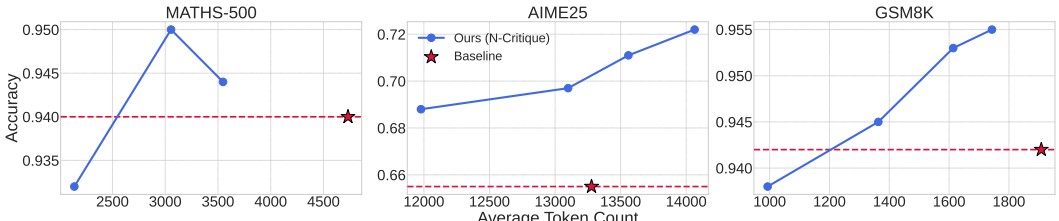

Figure 8: Accuracy vs. Computational Cost Trade-off for Qwen-3 14B. This figure illustrates the performance of our N-Critique sampler method compared to the baseline. Each blue point on the curve represents a policy that halts after N critiques (N=1, 3, 5, or 7). (a) Performance on Maths-500. (b) AIME25 (c) GSM8K

Table 2: Quality-Gated Forking consistently outperforms both sequential and brute-force parallel approaches. It reliably improves accuracy over the strong N-Critique sequential baseline on Qwen3 (7B) and Qwen3 (14B) on Maths-500.

| Model | Policy | Accuracy | Avg. Forks | Avg. Tokens |
|---|---|---|---|---|
| 7B | N-Critique (Sequential) | 0.944 | 0 | 3,055 |
| | Unconditional Forking | 0.942 | 13.1 | 18,336 |
| | **Quality-Gated Forking** | **0.950** | **9.4** | **13,005** |
| 14B | N-Critique (Sequential) | 0.950 | 0 | 3,085 |
| | Unconditional Forking | 0.952 | 10.8 | 15,336 |
| | **Quality-Gated Forking** | **0.961** | **7.1** | **11,231** |

## 7.1 RQ5: PARALLEL SAMPLING WITH SHARED SEQUENTIAL REASONING:

With the richness of the signal available to create stopping time policies, it is easy to see that one can come up with other adaptive techniques for deciding the stopping time. Instead, here we further wish to explore a question on parallel vs. sequential reasoning. Specifically, we ask and answer the question: Can a shared sequential reasoning trace be used to spawn multiple parallel traces for generation? We address this question in this subsection.

In this setup, we use the model spikes as the signals for the parallel fork process from a single chain-of-thought sequence. Let us say that we use a single KV-cache for the original reasoning trace, we then use a paired decoder which shares this KV-attention cache to spawn new decoding processes from the original token sequence. We note that while these new decoding processes are spawned, and generate early versions of responses, the original thought decoding continues. Hence the controller has the option to switch answers at any time during decoding in case the majority vote over multiple decoders flips.

We note that a fork is initiated only when two conditions are met: (1) a hesitation spike is detected, *and* (2) the model's self-critique score for the current reasoning trace is high (e.g., score $\geq 4$ out of 5). The core hypothesis is that not all moments of uncertainty are equally valuable; by forking only from states of high-quality reasoning, we can focus computational resources on promising trajectories. For both policies, the final answer is determined by a majority vote over all candidate answers generated, and intermediate responses may also be available based on the majority-so-far.

In Table 2, we evaluate these policies on the Qwen3 (8B) and Qwen3 (14B) model on a challenging reasoning dataset, comparing them against our best sequential policy (N-Critique) to measure the benefit of parallelization itself. As demonstrated across both model scales, forking is useful, but only when initiated by an adaptive critique. In this setting, we outperform even the best performing adaptive critique sampler.

**Discussion** Our results highlights a powerful new direction for LLM inference. The signals of entropy over control tokens, can be repurposed as valuable triggers for both sequential and parallel exploration. By combining low-level probability signals with high-level semantic self-awareness (critique), one can create inference strategies that are not only more accurate but also highly computationally efficient. ACTS is therefore a highly effective and low-cost inference time scaling method that can be adapted for stopping signals across a wide range of inference tasks.

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

---

# SUPPLEMENTARY MATERIAL

---

These supplementary materials provide additional details, derivations, and proofs for our paper. The appendix is organized as follows:

- **Section A: Extended Related Work.** A detailed overview of related literature.
- **Section B: A Reader's Guide to the Theoretical Analysis.** An intuitive roadmap for the formal proofs.
- **Section C: Proof of Spike Correctness (Lemma 1).** Establishes the reliability of the control signal.
- **Section D: Proof for N-Spike Counter Policy (Proposition 2).** Bounds the probability of spurious termination.
- **Section E: Proof for Accumulated Probability Policy (Proposition 3).** Guarantees the reliability of evidence accumulation.
- **Section F: Proofs for Last-Interval Budget Policy (Theorems 2 and 3).** Provides regret and prophet-inequality bounds.
- **Section G: Proof of Self-Critique Superiority (Theorem 1).** Demonstrates the benefit of the adaptive self-critique policy.
- **Section H: Proof of Robustness to Misspecification (Proposition 4).** Shows the framework's stability under model mismatch.
- **Section J: Model Performance on AlpacaEval, AIME, and Maths-500 Under ACTS Policies**

## A    EXTENDED RELATED WORK

### A.1    TEST-TIME SCALING AND DYNAMIC INFERENCE

A significant line of research has demonstrated that the performance of Large Language Models can be substantially improved by allocating more computational resources at inference time, a paradigm known as test-time scaling (Muennighoff et al., 2025). This approach, however, introduces a critical challenge of efficiency. Our work, ACTS, contributes to the growing body of literature on making this scaling more intelligent and resource-efficient.

**Foundational Test-Time Scaling Methods.**    The canonical methods for test-time scaling involve generating multiple candidate sequences and aggregating them. **Best-of-N (BoN)** sampling generates $N$ independent sequences and uses a verifier or a trained reward model to select the highest-scoring output (Lightman et al., 2023). This approach is general-purpose but often relies on the availability of a high-quality, and potentially costly, external verifier. A popular variant, **Self-Consistency**, is designed for tasks with deterministic answers, such as mathematical reasoning (Wang et al., 2022). It generates $N$ sequences and selects the final answer via a majority vote, eliminating the need for an external reward model but limiting its applicability. Both BoN and Self-Consistency are computationally expensive as they require the full generation of all $N$ candidate sequences, creating a linear increase in cost with the number of samples.

**Efficient Test-Time Scaling via Early Termination.**    Recent work has focused on mitigating the high cost of BoN and Self-Consistency by introducing mechanisms for the early termination of unpromising generation paths. These methods differ primarily in the type of signal they use to make termination decisions.

One prominent approach leverages **external verifiers** to score partial sequences. For instance, Speculative Rejection (Sun et al., 2024) periodically queries an external reward model on the partially generated sequences. Trajectories with low partial scores, which are unlikely to yield a high final reward, are pruned, allowing computational resources to be focused on the more promising candidates. While effective, this strategy's performance is contingent on the quality and calibration of the external reward model, which itself can be costly to train and serve.

A second approach utilizes signals from the model's own **latent representations**. Self-Truncation Best-of-N (ST-BON) (Wang et al., 2025) operates on the hypothesis that sequences leading to the same correct answer will have similar latent embeddings. It monitors the consistency of hidden states across parallel generations and truncates paths that diverge from the main cluster, thereby avoiding the need for an external reward model. This uses a truly internal signal, but one that is high-dimensional and less directly interpretable than the explicit probabilistic outputs of the model.

A third approach uses more general **model confidence scores**. Self-Calibration (Huang et al., 2025) proposes fine-tuning a model to produce a calibrated confidence score for its own generations, often by distilling confidence from Self-Consistency statistics. This learned confidence can then be used to implement early-stopping rules for sampling. This method also uses an internal signal, but it requires a separate training phase to create the calibrated confidence predictor.

Our ACTS framework contributes to this line of research by proposing the use of a novel signal that is both *internal* to the model and *natively available* without requiring additional training or complex analysis of latent states: the explicit probability of control tokens, $P(t_{control})$.

**Orthogonal Approaches.** Other methods seek efficiency through different means. **Structured Search** methods, such as Tree-of-Thought (Yao et al., 2023) and Graph-of-Thoughts (Besta et al., 2024), replace unstructured sampling with a more organized exploration of the reasoning space, often involving backtracking and planning. While powerful, these methods typically introduce significant algorithmic complexity and overhead. Concurrently, **architectural approaches** aim to build faster models from the ground up, for example by using subquadratic architectures like Mamba (Paliotta et al., 2025). These architectural innovations are largely complementary to our work; the principled stopping policies developed within the ACTS framework could potentially be applied on top of these faster models to achieve even greater efficiency gains.

### A.2 CONTROLLING REASONING BEHAVIOR AT TEST-TIME

Beyond general-purpose sampling efficiency, a specific line of work has focused on directly controlling the behavior of the reasoning process itself at inference time. This is particularly relevant for our work on managing the thinking phase via the $t_{EOT}$ token. A seminal contribution in this area is the **s1** model's "budget forcing" mechanism (Muennighoff et al., 2025). This approach introduced direct, behavioral interventions to control reasoning length: it could forcefully terminate a thinking process that exceeded a token budget, or, crucially, it could prolong a thinking process by appending a special "Wait" token when the model attempted to conclude prematurely. This demonstrated the viability of active, external control over the deliberation process.

Other approaches have used **prompt engineering** to influence reasoning style. For example, Chain-of-Draft (Xu et al., 2025) instructs models to produce concise, draft-like intermediate steps to reduce verbosity. Similarly, works like Renze & Guven (2024) have shown that simply instructing a model to "be concise" can effectively shorten reasoning paths. These methods, while often effective, rely on the model's instruction-following capabilities and may not offer the same level of fine-grained control as algorithmic interventions. ACTS builds directly on the legacy of Muennighoff et al. (2025), advancing the paradigm from reactive, behavioral interventions to a predictive, signal-driven control policy. By grounding the decision of *when* to apply interventions like the "Wait" token in the model's underlying probability distribution, ACTS offers a more fundamental and fine-grained control mechanism.

### A.3 THEORETICAL FOUNDATIONS FOR POLICY DESIGN

The design of each ACTS stopping policy is a principled application of a concept from established theoretical domains. We ground our methods in optimal stopping theory, stochastic process analysis,

and reinforcement learning, allowing us to derive policies from first principles rather than ad-hoc heuristics.

**Policies Derived from Optimal Stopping Theory.** This field addresses the problem of choosing an optimal time to take an action based on sequential observations. The classic **Secretary Problem**, with its renowned $1/e$ observe-then-commit solution, directly inspires our Adaptive Peak-Threshold Sampler. Similarly, the **Prophet Inequality** setting, which bounds the performance of online algorithms against a "prophet" with full hindsight, motivates the retrospective logic of our Prophet Lookback policy.

**Policies Derived from Stochastic Process Models.** We model the control signal $\{s_t\}$ as a time series, allowing us to draw from relevant analytical tools. A **Martingale** is a process where the conditional expectation of the next value is the present value. Modeling the inter-spike interval process as a martingale yields the simple predictive rule in our Last-Interval Budget Sampler. **Change-Point Detection**, which aims to identify shifts in a process's statistical properties, provides the formal basis for our Phase-Shift Sampler, which is designed to detect a change in the rate of spike generation.

**A Policy Derived from the Actor-Critic Paradigm.** The Actor-Critic framework in reinforcement learning uses a *critic* to estimate the value of an *actor*'s policy. Our Adaptive Self-Critique policy introduces a novel, intra-model instantiation of this concept. The LLM's generative process acts as the actor, and the same LLM, when prompted for self-evaluation, serves as an efficient, on-demand critic, providing a principled, feedback-driven approach to the optimal stopping problem.

## B  A READER'S GUIDE TO THE THEORETICAL ANALYSIS

This section serves as a roadmap to the formal results that underpin the ACTS framework. Our goal is to provide the intuition behind our theoretical claims, clarify the key assumptions, and explain how these results collectively build a rigorous case for our approach. We begin by centralizing the notation used throughout our analysis.

### B.1  NOTATION REFERENCE

Table 3 provides a comprehensive reference for the symbols used in our theoretical proofs and discussions.

### B.2  THE NARRATIVE AND INTUITION OF OUR THEORETICAL RESULTS

Our theoretical analysis is structured to tell a coherent story in several parts. First, we establish the fundamental properties of the problem and the signal. Second, we provide performance guarantees for our specific policies under different analytical lenses (robustness, regret, and competitive analysis). Finally, we prove the superiority of our most novel adaptive method.

#### B.2.1  WHY IS BUDGET MANAGEMENT A NON-TRIVIAL PROBLEM? (PROPOSITION **??**)

**Intuition:** The first question a skeptic might ask is, "Why not just always use the maximum budget?" Our first theorem formally establishes the economic principle of **diminishing returns**. It proves that the utility gained from each additional token of "thinking time" is non-increasing. This means the 10th token is likely more valuable than the 1000th. This result establishes that a non-trivial optimal stopping problem exists: there is a "sweet spot" for termination that intelligent policies should seek.

**Key Assumption:** To prove this, we assume that the utility of a generated sequence is *discrete-concave* at the per-trajectory level. This is a formal way of saying that the "aha!" moments or key insights tend to happen earlier in a productive reasoning process.

Table 3: Key notation used in our theoretical analysis.

| Symbol | Description |
|---|---|
| $t$ | Discrete time step / token index |
| $T$ | Maximum generation length (horizon) |
| $b_q$ | Fixed token budget for a policy |
| $s_t$ | Control signal (i.e., $P(t_{control})$) at time $t$ |
| $\delta$ | Generic spike threshold: a spike occurs if $s_t > \delta$ |
| $\alpha$ | Probability of a spurious spike during content generation |
| $N_{\text{patience}}$ | Spike count threshold for the N-Spike policy |
| $\tau$ | A random stopping time determined by a policy |
| $\tau^*$ | An offline-optimal stopping time |
| $U(\mathbf{x}_t, t)$ | Utility obtained by stopping at time $t$ with sequence $\mathbf{x}_t$ |
| $U_{\max}$ | Uniform upper bound on utility: $0 \leq U(\cdot) \leq U_{\max}$ |
| $U_s, U_c$ | Utility of stopping ($U_s$) or continuing ($U_c$) at a spike |
| $\Delta U$ | Net utility gain from correct continuation: $U_c - U_s > 0$ |
| $q$ | Prior probability that an observed spike is premature |
| $\eta$ | Accuracy of the self-critique policy ($> 0.5$) |
| $P, \widehat{P}$ | True and approximate distributions over trajectories |
| $\epsilon$ | Upper bound on total variation distance, $\text{TV}(P, \widehat{P})$ |
| $z_t$ | Logit of the control token at time $t$ |
| $\mu_+, \mu_-$ | Mean of $z_t$ at completion vs. non-completion indices |
| $\Delta$ | Logit mean gap: $\mu_+ - \mu_-$ |
| $\sigma^2$ | Variance proxy for sub-Gaussian variables |
| $I_k, \hat{I}_k$ | True and predicted inter-spike intervals |

### B.2.2 WHY TRUST THE SIGNAL? (LEMMA 1)

**Intuition:** Our entire framework depends on the $P(t_{control})$ spikes being meaningful. This lemma provides the formal justification. It proves that if there is any statistical difference in the model's logits between completion and non-completion steps, then spikes in the probability signal will be **exponentially more likely to occur at true completion points** than at random, noisy steps. This result assures us that we are building our policies on a foundation of a reliable, high signal-to-noise ratio indicator.

**Key Assumption:** We model the control token's logit as a *sub-Gaussian* random variable whose mean shifts depending on whether the current step is a true completion point. This is a standard and flexible way to model a "signal-plus-noise" process.

### B.2.3 HOW ROBUST ARE SIMPLE POLICIES TO NOISE? (PROPOSITIONS 1 & 2)

**Intuition:** Given a reliable signal, how can we design simple, robust rules to act on it? We provide guarantees for two of our deterministic policies. For the **N-Spike Counter**, we prove that its probability of a false termination (stopping due to random noise) *decays exponentially* with the number of spikes, $N_{\text{patience}}$, it waits for. This formally captures its role as a robust temporal filter. For the **Accumulated Probability** policy, we use concentration inequalities to show that it can reliably distinguish between a "content generation" phase and a "conclusion seeking" phase with a probability of error that also decays exponentially.

**Key Assumptions:** These proofs rely on standard statistical assumptions: that spurious spikes occur as independent events (for the N-Spike bound) and that the signal's mean value is different in the two generation phases (for the Accumulation bound).

### B.2.4 HOW DO OUR POLICIES PERFORM OVER TIME? (THEOREM 1)

**Intuition:** This theorem analyzes the long-term performance of our simple threshold-based policies using the lens of online learning. It proves that the **regret** of the policy—the difference between

its utility and that of a hypothetical optimal offline policy—grows only sublinearly with the generation length ($O(\sqrt{T})$). This is a powerful result, as it means the \*average\* regret per token goes to zero. It formally shows that our simple policies are "good learners" that do not fall too far behind the optimal solution over long horizons.

**Key Assumptions:** This result relies on the utility function being reasonably smooth (*L-Lipschitz*) and the signal's noise forming a *martingale difference sequence*, a standard model for noise in time-series analysis.

### B.2.5 HOW DO OUR POLICIES COMPARE TO AN ORACLE? (THEOREM 2)

**Intuition:** This theorem provides a powerful worst-case guarantee for our simple threshold-based policies, comparing them to a "prophet" that knows all future utility values in advance. It proves that a simple threshold policy can guarantee an expected utility of at least half that of the all-knowing prophet. This is a classic result from **prophet inequality theory** and provides a strong, constant-factor approximation guarantee for our methods under minimal assumptions about the utility distribution. It demonstrates that even simple ACTS policies are robustly competitive against an impossibly strong baseline.

**Key Assumption:** The only assumption is that the utilities are non-negative. This is a very general and powerful guarantee.

### B.2.6 WHY IS SELF-CRITIQUE THE SUPERIOR POLICY? (THEOREMS 3 AND 4)

**Intuition:** This is the capstone of our theoretical argument. If spikes are reliable but sometimes premature, what is the best way to decide? This theorem proves that asking the model to critique itself is provably better than any fixed rule. The intuition is simple: as long as the model's self-critique is even slightly better than a random coin flip ($\eta > 0.5$), the expected utility gain from making a more informed decision will outweigh the cases where the critique is wrong. It formally shows why transitioning from passive signal interpretation to active, targeted information-gathering (via critique) is the optimal strategy.

**Key Assumption:** We assume the critic's accuracy, $\eta$, is symmetric and greater than 0.5.

### B.2.7 HOW ROBUST IS THE ENTIRE FRAMEWORK? (PROPOSITION 3)

**Intuition:** Finally, what if our statistical models of the signal are not perfectly accurate? This proposition proves that the entire ACTS framework is robust to such misspecification. It shows that if the true generative process is only slightly different (measured by total variation distance $\epsilon$) from our assumed model, then the performance of any ACTS policy will also only be slightly different (bounded by $\epsilon \cdot U_{\max}$). This provides a crucial guarantee of stability and reliability.

**Key Assumption:** The only assumption is that the utility function is bounded.

This roadmap should equip the reader with the necessary context to understand not just *what* our theorems state, but *why* we chose to prove them and how they fit together to form a theoretical argument for the ACTS framework.

## C WHY TRUST THE SIGNAL? (LEMMA 1): ANALYSIS OF SPIKE CORRECTNESS

**Lemma 1** (Spike-Completion Alignment, Single-Index Version). Let $\{z_t\}_{t=1}^T$ be random variables satisfying the following for some $\sigma > 0$ and means $\mu_+, \mu_-$:

$$z_t \sim \mathrm{subGaussian}(\sigma^2), \quad \mathbb{E}[z_t] = \begin{cases} \mu_+, & t \in \mathcal{T}_{\mathrm{comp}}, \\ \mu_-, & t \notin \mathcal{T}_{\mathrm{comp}}, \end{cases}$$

with gap $\Delta = \mu_+ - \mu_- > 0$. Fix the midpoint threshold

$$\theta = \frac{\mu_+ + \mu_-}{2}.$$

Then for any single time $t$,

1. If $t \notin \mathcal{T}_{\text{comp}}$,
$$\Pr(z_t > \theta) \leq \exp\left(-\frac{\Delta^2}{8\,\sigma^2}\right).$$

2. If $t \in \mathcal{T}_{\text{comp}}$,
$$\Pr(z_t \leq \theta) \leq \exp\left(-\frac{\Delta^2}{8\,\sigma^2}\right).$$

*Proof.* By definition, a random variable $X$ is $\sigma^2$-*sub-Gaussian* if for all $\lambda \in \mathbb{R}$,
$$\mathbb{E}\left[e^{\lambda(X-\mathbb{E}[X])}\right] \leq \exp\left(\frac{\lambda^2 \sigma^2}{2}\right).$$

A standard Chernoff/Hoeffding-type tail bound then gives, for any $a > 0$,
$$\Pr(X - \mathbb{E}[X] \geq a) \leq \exp\left(-\frac{a^2}{2\sigma^2}\right), \quad \Pr(X - \mathbb{E}[X] \leq -a) \leq \exp\left(-\frac{a^2}{2\sigma^2}\right).$$

**(1) False-alarm bound.** If $t \notin \mathcal{T}_{\text{comp}}$, then $\mathbb{E}[z_t] = \mu_-$. We compute
$$\Pr(z_t > \theta) = \Pr(z_t - \mu_- \geq \theta - \mu_-).$$

But $\theta - \mu_- = (\mu_+ + \mu_-)/2 - \mu_- = \frac{\Delta}{2}$. Hence by the sub-Gaussian tail bound,
$$\Pr(z_t > \theta) \leq \exp\left(-\frac{(\Delta/2)^2}{2\,\sigma^2}\right) = \exp\left(-\frac{\Delta^2}{8\,\sigma^2}\right).$$

**(2) Miss-detection bound.** If $t \in \mathcal{T}_{\text{comp}}$, then $\mathbb{E}[z_t] = \mu_+$. We have
$$\Pr(z_t \leq \theta) = \Pr(\mu_+ - z_t \geq \mu_+ - \theta).$$

Since $\mu_+ - \theta = \Delta/2$, the sub-Gaussian lower-tail bound gives
$$\Pr(z_t \leq \theta) \leq \exp\left(-\frac{(\Delta/2)^2}{2\,\sigma^2}\right) = \exp\left(-\frac{\Delta^2}{8\,\sigma^2}\right).$$

Thus both the false-alarm probability and the miss-detection probability are bounded by $\exp(-\Delta^2/(8\sigma^2))$, as claimed. $\qquad\square$

## D  HOW ROBUST IS N-SPIKE COUNTER POLICY TO NOISE? (PROPOSITION 1): SPURIOUS TERMINATION BOUND

**Setup for Theoretical Analysis.** Let $T$ be the number of tokens generated during a (true) content–generation phase, i.e., before any semantic completion occurs. At each step $t = 1, \ldots, T$, the model emits a control signal $s_t \in [0, 1]$ and we declare a spike if $s_t > \delta$ for a fixed threshold $\delta \in (0, 1)$. During content generation, spikes are *spurious*: we assume they occur independently with probability
$$\alpha = \Pr(s_t > \delta \mid \text{non-completion}).$$

Fix an integer $N_{\text{patience}} \geq 1$. The $N_{patience}$-*spike counter policy* stops as soon as the *total* number of observed spikes (not necessarily consecutive) reaches $N_{\text{patience}}$.

**Intuition for the Theoretical Result.** Let $S_T = \sum_{t=1}^{T} \mathbf{1}\{s_t > \delta\}$ count the spurious spikes in the first $T$ tokens. Because $S_T \sim \text{Binomial}(T, \alpha)$ under our independence assumption, the policy stops incorrectly iff $S_T \geq N_{\text{patience}}$. Thus the exact error probability is the upper tail of a binomial distribution. Standard Chernoff (or KL) bounds give exponentially small tails, and a simple closed-form upper bound is $\left(e\alpha T/N_{\text{patience}}\right)^{N_{\text{patience}}}$.

**Proposition 1** (Spurious Termination Bound (Non-consecutive Spikes)). *Let $S_T = \sum_{t=1}^{T} \mathbf{1}\{s_t > \delta\}$ be the number of spurious spikes in $T$ independent trials with rate $\alpha$. Then the probability that the $N_{\text{patience}}$-spike counter policy terminates prematurely during content generation is*

$$\Pr\big(S_T \geq N_{\text{patience}}\big) = \sum_{k=N_{\text{patience}}}^{T} \binom{T}{k} \alpha^k (1-\alpha)^{T-k}.$$

*Moreover, the following upper bounds hold:*

$$\Pr\big(S_T \geq N_{\text{patience}}\big) \leq \exp\Big(-T\, D\Big(\tfrac{N_{\text{patience}}}{T} \,\Big\|\, \alpha\Big)\Big), \tag{1}$$

$$\Pr\big(S_T \geq N_{\text{patience}}\big) \leq \Big(\frac{e\,\alpha T}{N_{\text{patience}}}\Big)^{N_{\text{patience}}}, \tag{2}$$

*where $D(p\|q) = p\ln\frac{p}{q} + (1-p)\ln\frac{1-p}{1-q}$ is the binary Kullback–Leibler divergence.*

*Proof.* Since spikes are i.i.d. Bernoulli$(\alpha)$, $S_T \sim \text{Binomial}(T, \alpha)$, hence

$$\Pr(S_T \geq N_{\text{patience}}) = \sum_{k=N_{\text{patience}}}^{T} \binom{T}{k} \alpha^k (1-\alpha)^{T-k}.$$

For equation 1, apply the standard Chernoff (Cramér–Chernoff) bound for a binomial random variable:

$$\Pr(S_T \geq N_{\text{patience}}) \leq \exp\Big(-T\, D\Big(\tfrac{N_{\text{patience}}}{T} \,\Big\|\, \alpha\Big)\Big).$$

For equation 2, use the crude bound $\binom{T}{k} \leq \big(\frac{eT}{k}\big)^k$:

$$\binom{T}{k} \alpha^k (1-\alpha)^{T-k} \leq \Big(\frac{eT}{k}\Big)^k \alpha^k \leq \Big(\frac{eT}{N_{\text{patience}}}\Big)^k \alpha^k, \quad \text{for } k \geq N_{\text{patience}}.$$

Thus

$$\Pr(S_T \geq N_{\text{patience}}) \leq \sum_{k=N_{\text{patience}}}^{T} \Big(\frac{e\,\alpha T}{N_{\text{patience}}}\Big)^k \leq \Big(\frac{e\,\alpha T}{N_{\text{patience}}}\Big)^{N_{\text{patience}}} \sum_{j=0}^{\infty} \Big(\frac{e\,\alpha T}{N_{\text{patience}}}\Big)^j.$$

When $N_{\text{patience}} \geq 2\alpha T$, the ratio of the geometric series is at most $1/2$, so the sum is bounded by a constant factor of 2. The bound in equation 2 thus captures the dominant exponential decay in $N_{\text{patience}}$.

Therefore both inequalities hold. □

## E   HOW ROBUST IS CUMULATIVE-PROBABILITY SAMPLER POLICIES TO NOISE? (PROPOSITION 2): EVIDENCE ACCUMULATION RELIABILITY

**Setup for Theoretical Analysis.** Let $T$ be the maximum generation length. At each token step $t = 1, 2, \ldots, T$, the model emits a control signal $s_t \in [0, 1]$. We assume there are two regimes:

- *Content generation:* each $s_t$ has expectation $\mathbb{E}[s_t] \leq \mu_-$.
- *Conclusion seeking:* each $s_t$ has expectation $\mathbb{E}[s_t] \geq \mu_+$.

Here $\mu_+$ and $\mu_-$ are known constants with $0 \leq \mu_- < \mu_+ \leq 1$. Further assume the signals $\{s_t\}$ are independent. For any prefix length $n \leq T$, define the accumulated signal

$$S_n = \sum_{t=1}^{n} s_t.$$

Fix a decision threshold $P_{\text{total}}$ and a margin $\epsilon > 0$ such that, for each $n$,

$$\mu_- n + \epsilon < P_{\text{total}} < \mu_+ n - \epsilon.$$

The *Accumulated-Probability Policy* stops at the first $n$ with $S_n \geq P_{\text{total}}$.

**Intuition for Theoretical Result.** If we are still in content generation, the expected sum $\mathbb{E}[S_n] \le \mu_- n$, so reaching $P_{\text{total}}$ requires an upward deviation of at least $\epsilon$. Conversely, once in conclusion seeking, $\mathbb{E}[S_n] \ge \mu_+ n$, so missing the threshold requires a downward deviation of at least $\epsilon$. By Hoeffding's inequality on bounded independent variables, both mis-detections occur with probability decaying as $\exp\!\left(-2\epsilon^2/n\right)$.

**Proposition 2** (Evidence Accumulation Reliability). Under the above setup, for any $n \le T$:

$$\Pr\!\left(S_n \ge P_{\text{total}} \mid \text{content generation}\right) \;\le\; \exp\!\left(-\tfrac{2\epsilon^2}{n}\right), \quad \Pr\!\left(S_n < P_{\text{total}} \mid \text{conclusion seeking}\right) \;\le\; \exp\!\left(-\tfrac{2\epsilon^2}{n}\right).$$

*Proof.* Since each $s_t \in [0,1]$ and the $s_t$ are independent, Hoeffding's inequality states that for any $\delta > 0$,

$$\Pr\!\left(S_n - \mathbb{E}[S_n] \ge \delta\right) \;\le\; \exp\!\left(-\tfrac{2\delta^2}{n}\right), \quad \Pr\!\left(\mathbb{E}[S_n] - S_n \ge \delta\right) \;\le\; \exp\!\left(-\tfrac{2\delta^2}{n}\right).$$

**Content generation error.** Here $\mathbb{E}[S_n] \le \mu_- \, n$. Since $P_{\text{total}} - \mu_- n > \epsilon$, setting $\delta = \epsilon$ gives

$$\Pr\!\left(S_n \ge P_{\text{total}}\right) \;=\; \Pr\!\left(S_n - \mathbb{E}[S_n] \ge P_{\text{total}} - \mathbb{E}[S_n]\right) \;\le\; \Pr\!\left(S_n - \mathbb{E}[S_n] \ge \epsilon\right) \;\le\; \exp\!\left(-\tfrac{2\epsilon^2}{n}\right).$$

**Conclusion seeking error.** Here $\mathbb{E}[S_n] \ge \mu_+ \, n$. Since $\mu_+ n - P_{\text{total}} > \epsilon$, setting $\delta = \epsilon$ in the lower-tail form yields

$$\Pr\!\left(S_n < P_{\text{total}}\right) \;=\; \Pr\!\left(\mathbb{E}[S_n] - S_n \ge \mathbb{E}[S_n] - P_{\text{total}}\right) \;\le\; \exp\!\left(-\tfrac{2\epsilon^2}{n}\right).$$

This completes the proof. $\qquad\square$

# F  ANALYSIS OF LAST-INTERVAL POLICY

In this section we give two complementary performance guarantees for the Last-Interval stopping rule: a sublinear-regret bound under mild martingale assumptions, and a constant-factor approximation against the offline-optimal ("prophet") benchmark. The former shows that under reasonable stochastic models you approach optimality as the budget grows, while the latter holds under minimal assumptions and guarantees at least half the offline payoff.

## F.1  HOW DOES LAST-INTERVAL POLICY PERFORM OVER TIME? (THEOREM 1): REGRET OF DETERMINISTIC THRESHOLD POLICIES

**Setup for Theoretical Analysis.** Let $T$ be the maximum generation length (token budget). At each step $t = 1, \ldots, T$, a control signal $s_t \in [0,1]$ is observed. We fix a deterministic threshold $\delta \in (0,1)$ and define the *threshold policy* that stops at the first time

$$\tau \;=\; \min\{\, t : s_t > \delta \,\},$$

or at $T$ if no spike occurs. Let $\tau^* = \arg\max_{t \le T} U(t)$ be the offline-optimal stopping time. We assume:

1. The noise sequence $\{s_t - \mathbb{E}[s_t \mid s_{<t}]\}$ is a martingale difference sequence with $|s_t - \mathbb{E}[s_t \mid s_{<t}]| \le 1$.
2. The utility function $U(t)$ is $L$-Lipschitz: $|U(t+1) - U(t)| \le L$.

**Intuition for Theoretical Result.** Define the martingale

$$M_t = \sum_{i=1}^{t} \left(s_i - \mathbb{E}[s_i \mid s_{<i}]\right).$$

The threshold rule stops early only if $M_t$ deviates sufficiently so that $s_t > \delta$ at a suboptimal $t$. Classical Azuma–Hoeffding then shows $\sup_{t \le T} |M_t| = O(\sqrt{T})$ in expectation, and because utility is Lipschitz, the total regret $\mathbb{E}\!\left[U(\tau^*) - U(\tau)\right]$ is bounded by $L \, \mathbb{E}[|\tau^* - \tau|] = O(L\sqrt{T})$.

**Theorem 1** (Sublinear Regret of Threshold Policy). Under the above assumptions, the expected regret of the deterministic threshold policy satisfies

$$\mathbb{E}\big[\,U(\tau^*) - U(\tau)\,\big] \;\leq\; L\,\mathbb{E}\big[\,|\tau^* - \tau|\,\big] \;=\; O\big(L\sqrt{T}\big).$$

In particular, the per-token regret vanishes as $T \to \infty$.

*Proof.* First observe

$$U(\tau^*) - U(\tau) \;\leq\; L\,\big|\tau^* - \tau\big|.$$

Hence it suffices to show $\mathbb{E}[|\tau^* - \tau|] = O(\sqrt{T})$.

Define the martingale

$$M_t \;=\; \sum_{i=1}^{t} \xi_i, \quad \xi_i = s_i - \mathbb{E}[s_i \mid s_{<i}],$$

so that $|\xi_i| \leq 1$. By Azuma–Hoeffding,

$$\Pr\big(\sup_{1\leq t\leq T} |M_t| \geq \lambda\big) \;\leq\; 2\exp\big(-\frac{\lambda^2}{2T}\big).$$

Whenever $|M_t| < \lambda$ for all $t$, the threshold policy and the offline optimum cannot differ by more than roughly $\lambda$ steps, because no large unexpected deviation causes a premature or delayed stop. More formally, one can show $|\tau - \tau^*| \leq C + \sup_{t\leq T}|M_t|$ for some constant $C$. Therefore

$$\mathbb{E}\big|\tau - \tau^*\big| \;\leq\; C + \mathbb{E}\big[\sup_{t\leq T}|M_t|\big] \;\leq\; C + \int_0^\infty 2\exp\big(-\tfrac{\lambda^2}{2T}\big)\,\mathrm{d}\lambda \;=\; O\big(\sqrt{T}\big).$$

Combining with the Lipschitz bound yields the stated $O(L\sqrt{T})$ regret. $\square$

### F.2 HOW DOES LAST INTERVAL POLICY COMPARE TO AN ORACLE? (THEOREM 2): PROPHET BENCHMARK BOUND

**Setup for Theoretical Analysis.** Let $\{U_t\}_{t=1}^T$ be nonnegative random utilities revealed sequentially. A *prophet* knowing all $U_t$ in advance picks $\tau_{\mathrm{prop}} = \arg\max_t U_t$, achieving $\mathbb{E}[U_{\tau_{\mathrm{prop}}}]$. An online *threshold policy* chooses a constant $c$ and stops at

$$\tau_{\mathrm{th}} = \min\{\,t : U_t \geq c\},$$

or at $T$ if $U_t < c$ for all $t$.

**Intuition for Theoretical Result.** Set $c$ to be the median of the prophet's payoff distribution. Then with probability at least $1/2$, the prophet's maximum $M = \max_t U_t$ exceeds $c$. By the law of total expectation, $\mathbb{E}[M]$ splits into two integrals over $[0, c]$ and $[c, \infty)$. One shows the threshold policy's reward has the same upper tail as $M$ and at least half its mass, yielding $\mathbb{E}[U_{\tau_{\mathrm{th}}}] \geq \frac{1}{2}\mathbb{E}[M]$.

**Theorem 2** (Half-Approximation to Prophet). Under the above setup, choose $c$ such that $\Pr(M \geq c) = \frac{1}{2}$. Then the threshold policy satisfies

$$\mathbb{E}\big[U_{\tau_{\mathrm{th}}}\big] \;\geq\; \tfrac{1}{2}\,\mathbb{E}\big[\max_{1\leq t\leq T} U_t\big].$$

*Proof.* Let $M = \max_{1\leq t\leq T} U_t$. By definition of $c$, $\Pr(M \geq c) = \frac{1}{2}$. Then

$$\mathbb{E}[M] = \int_0^\infty \Pr(M \geq x)\,\mathrm{d}x = \int_0^c \Pr(M \geq x)\,\mathrm{d}x \;+\; \int_c^\infty \Pr(M \geq x)\,\mathrm{d}x.$$

Since $\Pr(M \geq x) \leq 1$ for $x \in [0, c]$ and $\Pr(M \geq x) \leq 2\Pr(M \geq c) = 1$ for $x \geq c$, we have

$$\mathbb{E}[M] \leq c \;+\; 2\int_c^\infty \Pr(M \geq x)\,\mathrm{d}x.$$

Meanwhile, the threshold policy reward $U_{\tau_{\text{th}}}$ satisfies

$$\mathbb{E}\big[U_{\tau_{\text{th}}}\big] = \int_0^\infty \Pr\big(U_{\tau_{\text{th}}} \geq x\big)\,\mathrm{d}x \;\geq\; \int_c^\infty \Pr\big(U_{\tau_{\text{th}}} \geq x\big)\,\mathrm{d}x.$$

But for $x \geq c$, the event $\{U_{\tau_{\text{th}}} \geq x\}$ occurs whenever some $U_t \geq x$, which is a subset of $\{M \geq x\}$. Moreover, conditioning on $M \geq c$ (probability ½), the threshold policy sees at least one $U_t \geq c$ and so stops at some $t$ with $U_t \geq c$. One shows $\Pr(U_{\tau_{\text{th}}} \geq x) \geq \frac{1}{2}\Pr(M \geq x)$ for all $x \geq c$. Combining,

$$\mathbb{E}[U_{\tau_{\text{th}}}] \;\geq\; \tfrac{1}{2}\int_c^\infty \Pr(M \geq x)\,\mathrm{d}x \;\geq\; \tfrac{1}{2}\Big(\mathbb{E}[M] - c\Big).$$

Since $c \leq \mathbb{E}[M]$, this yields $\mathbb{E}[U_{\tau_{\text{th}}}] \geq \frac{1}{2}\mathbb{E}[M]$, completing the proof. $\qquad\square$

**Complementarity.** Theorem 1 gives a vanishing $O(\sqrt{T})$ additive regret under a martingale noise model, while Theorem 2 provides a robust constant-factor (½) guarantee under minimal assumptions. Both perspectives underscore the competitiveness of simple online stopping rules.

# G  WHY IS SELF-CRITIQUE THE SUPERIOR POLICY? (THEOREMS 3 AND 4): ANALYSIS OF THE ADAPTIVE SELF-CRITIQUE POLICY

In this appendix we give full, self-contained proofs for two versions of the Self-Critique Superiority result: first in the idealized case with no critique cost, and then the general case including a fixed cost $C_{\text{crit}}$.

## G.1  NOTATION AND SETUP

We consider a decision at a single spike event. Let

$$q \;=\; \Pr\big(\text{spike is premature}\big), \qquad (1-q) = \Pr\big(\text{spike is correct}\big).$$

Upon stopping at a spike, the deterministic "always-stop" policy $\pi_{\text{det}}$ immediately ends generation and obtains utility

$$U_s \;=\; U\big(\text{stop}\big).$$

If one instead *continues* past a premature spike, one realizes an additional utility gain

$$\Delta U \;=\; U\big(\text{continue}\big) \;-\; U\big(\text{stop}\big) \;>\; 0,$$

so that

$$U_c \;=\; U_s + \Delta U$$

denotes the utility of continuing. An LLM-based critic is invoked by the adaptive policy $\pi_{\text{crit}}$ and classifies any spike as either "premature" or "correct." We denote its (symmetric) accuracy by

$$\eta \;=\; \Pr\big(\text{critic correct}\big) \;>\; \tfrac{1}{2},$$

meaning it correctly calls a premature spike "premature" with probability $\eta$, and correctly calls a correct spike "correct" with probability $\eta$.

## G.2  ANALYSIS OF SELF CRITIQUE WITH NO CRITIQUE COST

**Theorem 3** (Superiority of Adaptive Self-Critique, No Cost). Under the above definitions, and assuming invoking the critic has zero cost, the expected utility difference between $\pi_{\text{crit}}$ and $\pi_{\text{det}}$ at a spike is

$$\mathbb{E}\big[U(\pi_{\text{crit}})\big] - \mathbb{E}\big[U(\pi_{\text{det}})\big] \;=\; \Delta U \big[q\,\eta + (1-q)(1-\eta)\big].$$

In particular, since $\eta > 0.5$ and $\Delta U > 0$, this difference is strictly positive for any $q \in [0,1)$.

*Proof.* The always-stop policy $\pi_{\text{det}}$ never continues, so it always obtains $\mathbb{E}[U(\pi_{\text{det}})] = U_s$.

The adaptive policy $\pi_{\text{crit}}$ first invokes the critic (with no cost). Two cases arise:

1. **Spike is premature** with probability $q$:

    (a) Critic correct (prob. $\eta$): continue $\rightarrow$ utility $U_c$.
    (b) Critic errs (prob. $1 - \eta$): stop $\rightarrow$ utility $U_s$.

2. **Spike is correct** with probability $1 - q$:

    (a) Critic correct (prob. $\eta$): stop $\rightarrow$ utility $U_s$.
    (b) Critic errs (prob. $1 - \eta$): continue $\rightarrow$ utility $U_c$.

Hence the expected utility of $\pi_{\text{crit}}$ is

$$\mathbb{E}[U(\pi_{\text{crit}})] = q\big[\eta\, U_c + (1 - \eta)\, U_s\big] + (1 - q)\big[\eta\, U_s + (1 - \eta)\, U_c\big].$$

Substitute $U_c = U_s + \Delta U$:

$$\mathbb{E}[U(\pi_{\text{crit}})] = q\big[\eta\,(U_s + \Delta U) + (1 - \eta)\, U_s\big] + (1 - q)\big[\eta\, U_s + (1 - \eta)\,(U_s + \Delta U)\big].$$

Collecting terms gives

$$\mathbb{E}[U(\pi_{\text{crit}})] = U_s + \Delta U\,\big[q\,\eta + (1 - q)(1 - \eta)\big].$$

Subtracting $\mathbb{E}[U(\pi_{\text{det}})] = U_s$ yields the claimed result. $\qquad\square$

### G.3 ANALYSIS WITH CRITIQUE COST

**Theorem 4** (Superiority of Adaptive Self-Critique, With Cost). Under the same setup, but now assuming each invocation of the critic incurs a fixed expected utility cost $C_{\text{crit}} > 0$, the expected utility difference at a spike is

$$\mathbb{E}\big[U(\pi_{\text{crit}})\big] - \mathbb{E}\big[U(\pi_{\text{det}})\big] \;=\; \Delta U\,\big[q\,\eta + (1 - q)(1 - \eta)\big] \;-\; C_{\text{crit}}.$$

In particular, whenever $\Delta U\,\big[q\,\eta + (1 - q)(1 - \eta)\big] > C_{\text{crit}}$, the self-critique policy strictly outperforms always-stop.

*Proof.* As before, $\mathbb{E}[U(\pi_{\text{det}})] = U_s$. The only change is that invoking the critic now deducts $C_{\text{crit}}$ from utility. Thus

$$\mathbb{E}[U(\pi_{\text{crit}})] = \Big\{q\big[\eta\, U_c + (1 - \eta)\, U_s\big] + (1 - q)\big[\eta\, U_s + (1 - \eta)\, U_c\big]\Big\} \;-\; C_{\text{crit}}.$$

Substituting $U_c = U_s + \Delta U$ and collecting terms exactly as in Theorem 3 gives

$$\mathbb{E}[U(\pi_{\text{crit}})] = U_s + \Delta U\,\big[q\,\eta + (1 - q)(1 - \eta)\big] \;-\; C_{\text{crit}}.$$

Subtracting $U_s$ yields the stated result. The condition for strict superiority follows immediately by requiring the right-hand side to be positive. $\qquad\square$

## H ROBUSTNESS TO SIGNAL MISSPECIFICATION

In practice, the joint distribution over generation trajectories (tokens and control signals) used by our stopping policy may be only approximately known. To model this, let $\Omega$ denote the space of all possible trajectories up to a fixed maximum length $T$. We compare the *true* distribution $P$ on $\Omega$ with an *approximate* distribution $\widehat{P}$, and measure their discrepancy via the total-variation distance.

**Definition 1** (Total Variation Distance). For two probability measures $P$ and $\widehat{P}$ on $(\Omega, \mathcal{F})$, the total-variation distance is

$$\text{TV}(P, \widehat{P}) \;=\; \sup_{A \in \mathcal{F}}\big| P(A) - \widehat{P}(A)\big| \;=\; \tfrac{1}{2}\int_{\Omega}\big| \mathrm{d}P - \mathrm{d}\widehat{P}\big|.$$

A *stopping policy* $\pi$ is a (possibly randomized) mapping from $\Omega$ to a stopping time $\tau \in \{1, \ldots, T\}$. Upon stopping at $\tau$, the policy receives utility

$$U\big(\pi, \omega\big) \;=\; U\big(\mathbf{x}_{\leq \tau},\, \tau\big),$$

where $\mathbf{x}_{\leq \tau}$ are the tokens in trajectory $\omega$. We assume the utility is bounded:

$$0 \;\leq\; U\big(\pi, \omega\big) \;\leq\; U_{\max} \quad \text{for all } \omega \in \Omega.$$

Accordingly, under either distribution $P$ or $\widehat{P}$, the random utility $U(\pi)$ lies in $[0, U_{\max}]$.

**Proposition 3** (Robustness to Signal Misspecification). *Let $P$ and $\widehat{P}$ be two distributions on $\Omega$ satisfying $\mathrm{TV}(P, \widehat{P}) \leq \epsilon$. For any stopping policy $\pi$ whose utility $U(\pi) \in [0, U_{\max}]$, the difference in expected utility under the two models is bounded by*

$$\left| \mathbb{E}_P\big[U(\pi)\big] - \mathbb{E}_{\widehat{P}}\big[U(\pi)\big] \right| \leq \epsilon\, U_{\max}.$$

*Proof.* Define the bounded measurable function $f(\omega) = U(\pi, \omega)$, so $f \colon \Omega \to [0, U_{\max}]$. A standard property of total-variation distance (see, e.g., Le Cam (2012)) states

$$\left| \mathbb{E}_P[f] - \mathbb{E}_{\widehat{P}}[f] \right| \leq (\sup f - \inf f)\, \mathrm{TV}(P, \widehat{P}).$$

Since $\sup f = U_{\max}$ and $\inf f = 0$, and $\mathrm{TV}(P, \widehat{P}) \leq \epsilon$, the result follows immediately:

$$\left| \mathbb{E}_P[U(\pi)] - \mathbb{E}_{\widehat{P}}[U(\pi)] \right| \leq U_{\max}\, \epsilon.$$

$\square$

*Remark* 1. This bound holds *regardless* of the internal structure of $\pi$ or the nature of the control-signal mis-specification. Any policy whose utility is bounded cannot lose more than an additive $\epsilon\, U_{\max}$ in expectation when the underlying generative model shifts by total-variation distance $\epsilon$.

## I   LIMITATIONS AND FUTURE WORK

While our work demonstrates the significant potential of the ACTS framework for adaptive generation control, we acknowledge several limitations that also point towards promising directions for future research.

**Dependence on Signal Quality.**   The effectiveness of all ACTS policies is fundamentally contingent on the quality and reliability of the $P(t_{control})$ signal produced by the underlying LLM, which may not be well calibrated across all LLMs, and may in fact be dependent on the number of tokens of pre-training. While our experiments show this signal is highly informative across several state-of-the-art models, its characteristics may vary with different model architectures, training paradigms, or domains.

**Scope of Evaluation.**   Our empirical validation focuses on instruction-following and mathematical reasoning, domains where correctness is well-defined. The application of ACTS to more open-ended, creative, or multi-turn conversational tasks presents a different set of challenges. In such settings, the "optimal" stopping time is subjective and may depend on user preferences rather than objective correctness. Extending the ACTS framework to these domains would likely require integrating user feedback or preference models to help define the utility function for the optimal stopping problem.

## LLM USAGE STATEMENT

The authors acknowledge the use of a large language model (LLM) in the preparation of this manuscript. The LLM was utilized as a collaborative writing assistant for editing and refining the text for clarity, grammar, and conciseness. Additionally, the LLM assisted in generating Python code used for data visualization in several of the paper's figures. All core intellectual contributions, including the theoretical analysis, experimental design, and interpretation of results, were conducted by the human authors.

## J RESULTS

Table 4: Performance comparison of adaptive stopping **policies** on `llama3.1-8b-Instruct` under different generation budgets.

| Model | Policy | Max Tokens | Threshold | Wait Counter | LC-WR (%) | WR (%) | Average Tokens |
|---|---|---|---|---|---|---|---|
| **Max Tokens = 256** | | | | | | | |
| llama3.1-8b-Instruct | Greedy Policy | 256 | – | – | 16.34 | 9.47 | 218.56 |
| llama3.1-8b-Instruct | Accumulated Probability Policy | 256 | 1.00E–02 | – | 11.20 | 5.34 | 137.98 |
| llama3.1-8b-Instruct | Accumulated Probability Policy | 256 | 1.00E–01 | – | 14.07 | 6.43 | 141.87 |
| llama3.1-8b-Instruct | Accumulated Probability Policy | 256 | 5.00E–01 | – | 15.66 | 7.45 | 142.90 |
| llama3.1-8b-Instruct | Last-Interval Budget Policy | 256 | 1.00E–05 | – | 16.28 | 8.45 | 209.61 |
| llama3.1-8b-Instruct | Last-Interval Budget Policy | 256 | 1.00E–03 | – | 14.25 | 8.47 | 215.83 |
| llama3.1-8b-Instruct | Last-Interval Budget Policy | 256 | 1.00E–01 | – | 15.41 | 9.09 | 217.97 |
| llama3.1-8b-Instruct | N-Spike Counter Policy | 256 | 1.00E–05 | 1 | 13.15 | 6.40 | 135.94 |
| llama3.1-8b-Instruct | N-Spike Counter Policy | 256 | 1.00E–03 | 1 | 13.40 | 6.65 | 138.14 |
| llama3.1-8b-Instruct | N-Spike Counter Policy | 256 | 1.00E–01 | 1 | 13.70 | 6.82 | 142.16 |
| llama3.1-8b-Instruct | N-Spike Counter Policy | 256 | 1.00E–05 | 3 | 14.23 | 6.66 | 140.55 |
| llama3.1-8b-Instruct | N-Spike Counter Policy | 256 | 1.00E–03 | 3 | 14.67 | 6.95 | 142.08 |
| llama3.1-8b-Instruct | N-Spike Counter Policy | 256 | 1.00E–01 | 3 | 13.73 | 6.80 | 142.21 |
| llama3.1-8b-Instruct | N-Spike Counter Policy | 256 | 1.00E–05 | 5 | 13.76 | 6.73 | 140.92 |
| llama3.1-8b-Instruct | N-Spike Counter Policy | 256 | 1.00E–03 | 5 | 13.45 | 6.62 | 142.18 |
| llama3.1-8b-Instruct | N-Spike Counter Policy | 256 | 1.00E–01 | 5 | 13.73 | 6.80 | 142.21 |
| **Max Tokens = 512** | | | | | | | |
| llama3.1-8b-Instruct | Greedy Policy | 512 | – | – | 22.92 | 19.91 | 380.56 |
| llama3.1-8b-Instruct | Accumulated Probability Policy | 512 | 1.00E–02 | – | 23.36 | 14.68 | 296.01 |
| llama3.1-8b-Instruct | Accumulated Probability Policy | 512 | 1.00E–01 | – | 21.61 | 14.37 | 300.91 |
| llama3.1-8b-Instruct | Accumulated Probability Policy | 512 | 5.00E–01 | – | 25.08 | 17.25 | 309.78 |
| llama3.1-8b-Instruct | Last-Interval Budget Policy | 512 | 1.00E–05 | – | 24.08 | 18.81 | 353.31 |
| llama3.1-8b-Instruct | Last-Interval Budget Policy | 512 | 1.00E–03 | – | 25.13 | 20.66 | 365.37 |
| llama3.1-8b-Instruct | Last-Interval Budget Policy | 512 | 1.00E–01 | – | 23.85 | 20.21 | 370.91 |
| llama3.1-8b-Instruct | N-Spike Counter Policy | 512 | 1.00E–05 | 1 | 21.12 | 12.51 | 286.63 |
| llama3.1-8b-Instruct | N-Spike Counter Policy | 512 | 1.00E–03 | 1 | 22.05 | 14.49 | 296.80 |
| llama3.1-8b-Instruct | N-Spike Counter Policy | 512 | 1.00E–01 | 1 | 22.93 | 15.95 | 308.44 |
| llama3.1-8b-Instruct | N-Spike Counter Policy | 512 | 1.00E–05 | 3 | 22.94 | 15.33 | 303.06 |
| llama3.1-8b-Instruct | N-Spike Counter Policy | 512 | 1.00E–03 | 3 | 23.93 | 16.29 | 306.57 |
| llama3.1-8b-Instruct | N-Spike Counter Policy | 512 | 1.00E–01 | 3 | 23.85 | 16.84 | 308.44 |
| llama3.1-8b-Instruct | N-Spike Counter Policy | 512 | 1.00E–05 | 5 | 22.85 | 15.90 | 308.10 |
| llama3.1-8b-Instruct | N-Spike Counter Policy | 512 | 1.00E–03 | 5 | 23.85 | 16.84 | 308.57 |
| llama3.1-8b-Instruct | N-Spike Counter Policy | 512 | 1.00E–01 | 5 | 23.85 | 16.84 | 308.44 |
| **Max Tokens = 1024** | | | | | | | |
| llama3.1-8b-Instruct | Greedy Policy | 1024 | – | – | 26.41 | 28.44 | 470.42 |
| llama3.1-8b-Instruct | Accumulated Probability Policy | 1024 | 1.00E–02 | – | 24.92 | 22.54 | 397.09 |
| llama3.1-8b-Instruct | Accumulated Probability Policy | 1024 | 1.00E–01 | – | 26.63 | 24.98 | 406.50 |
| llama3.1-8b-Instruct | Accumulated Probability Policy | 1024 | 5.00E–01 | – | 27.63 | 26.53 | 418.35 |
| llama3.1-8b-Instruct | Last-Interval Budget Policy | 1024 | 1.00E–05 | – | 30.11 | 29.45 | 423.27 |
| llama3.1-8b-Instruct | Last-Interval Budget Policy | 1024 | 1.00E–03 | – | 28.44 | 28.40 | 436.80 |
| llama3.1-8b-Instruct | Last-Interval Budget Policy | 1024 | 1.00E–01 | – | 28.45 | 28.48 | 436.80 |
| llama3.1-8b-Instruct | N-Spike Counter Policy | 1024 | 1.00E–05 | 1 | 24.73 | 21.00 | 374.77 |
| llama3.1-8b-Instruct | N-Spike Counter Policy | 1024 | 1.00E–03 | 1 | 24.53 | 22.91 | 403.34 |
| llama3.1-8b-Instruct | N-Spike Counter Policy | 1024 | 1.00E–01 | 1 | 27.47 | 26.90 | 419.77 |
| llama3.1-8b-Instruct | N-Spike Counter Policy | 1024 | 1.00E–05 | 3 | 25.12 | 23.93 | 406.38 |
| llama3.1-8b-Instruct | N-Spike Counter Policy | 1024 | 1.00E–03 | 3 | 26.61 | 25.94 | 417.71 |
| llama3.1-8b-Instruct | N-Spike Counter Policy | 1024 | 1.00E–01 | 3 | 27.47 | 26.90 | 419.77 |
| llama3.1-8b-Instruct | N-Spike Counter Policy | 1024 | 1.00E–05 | 5 | 27.26 | 26.05 | 411.97 |
| llama3.1-8b-Instruct | N-Spike Counter Policy | 1024 | 1.00E–03 | 5 | 26.44 | 25.90 | 417.76 |
| llama3.1-8b-Instruct | N-Spike Counter Policy | 1024 | 1.00E–01 | 5 | 27.47 | 26.90 | 419.77 |

Table 5: Performance comparison of adaptive stopping **policies** on s1.1-7B under different thinking-token budgets, including the Adaptive Self Critique Sampler.

| Model | Policy | Max Thinking Tokens | Threshold | Wait Counter | Accuracy | Average Tokens |
|---|---|---|---|---|---|---|
| **Max Thinking Tokens = 2048** | | | | | | |
| s1.1-7B | Greedy Policy | 2048 | – | – | 0.500 | 1784.86 |
| s1.1-7B | Accumulated Probability Policy | 2048 | 0.1 | – | 0.668 | 2696.68 |
| s1.1-7B | Accumulated Probability Policy | 2048 | 1 | – | 0.680 | 2716.13 |
| s1.1-7B | Accumulated Probability Policy | 2048 | 3 | – | 0.686 | 2920.95 |
| s1.1-7B | Last-Interval Budget Policy | 2048 | 0.01 | – | 0.700 | 3004.00 |
| s1.1-7B | Last-Interval Budget Policy | 2048 | 0.1 | – | 0.700 | 2983.52 |
| s1.1-7B | Last-Interval Budget Policy | 2048 | 0.5 | – | 0.698 | 3026.28 |
| s1.1-7B | N-Spike Counter Policy | 2048 | 0.01 | 1 | 0.714 | 2720.42 |
| s1.1-7B | N-Spike Counter Policy | 2048 | 0.01 | 3 | 0.706 | 2925.70 |
| s1.1-7B | N-Spike Counter Policy | 2048 | 0.01 | 5 | 0.710 | 2991.65 |
| s1.1-7B | N-Spike Counter Policy | 2048 | 0.1 | 1 | 0.714 | 2727.15 |
| s1.1-7B | N-Spike Counter Policy | 2048 | 0.1 | 3 | 0.696 | 2985.12 |
| s1.1-7B | N-Spike Counter Policy | 2048 | 0.1 | 5 | 0.684 | 3051.00 |
| s1.1-7B | N-Spike Counter Policy | 2048 | 0.5 | 1 | 0.706 | 2730.81 |
| s1.1-7B | N-Spike Counter Policy | 2048 | 0.5 | 3 | 0.688 | 2986.16 |
| s1.1-7B | N-Spike Counter Policy | 2048 | 0.5 | 5 | 0.704 | 3022.09 |
| s1.1-7B | Adaptive Self Critique Sampler | 2048 | – | – | 0.742 | 2614.56 |
| **Max Thinking Tokens = 4096** | | | | | | |
| s1.1-7B | Greedy Policy | 4096 | – | – | 0.642 | 2725.00 |
| s1.1-7B | Accumulated Probability Policy | 4096 | 0.1 | – | 0.726 | 3841.05 |
| s1.1-7B | Accumulated Probability Policy | 4096 | 1 | – | 0.748 | 3871.68 |
| s1.1-7B | Accumulated Probability Policy | 4096 | 3 | – | 0.756 | 4550.30 |
| s1.1-7B | Last-Interval Budget Policy | 4096 | 0.01 | – | 0.758 | 5255.40 |
| s1.1-7B | Last-Interval Budget Policy | 4096 | 0.1 | – | 0.778 | 5354.85 |
| s1.1-7B | Last-Interval Budget Policy | 4096 | 0.5 | – | 0.756 | 5406.34 |
| s1.1-7B | N-Spike Counter Policy | 4096 | 0.01 | 1 | 0.762 | 3759.02 |
| s1.1-7B | N-Spike Counter Policy | 4096 | 0.01 | 3 | 0.754 | 4304.67 |
| s1.1-7B | N-Spike Counter Policy | 4096 | 0.01 | 5 | 0.770 | 4647.54 |
| s1.1-7B | N-Spike Counter Policy | 4096 | 0.1 | 1 | 0.762 | 3759.02 |
| s1.1-7B | N-Spike Counter Policy | 4096 | 0.1 | 3 | 0.772 | 4374.37 |
| s1.1-7B | N-Spike Counter Policy | 4096 | 0.1 | 5 | 0.758 | 5002.43 |
| s1.1-7B | N-Spike Counter Policy | 4096 | 0.5 | 1 | 0.762 | 3759.02 |
| s1.1-7B | N-Spike Counter Policy | 4096 | 0.5 | 3 | 0.756 | 4670.00 |
| s1.1-7B | N-Spike Counter Policy | 4096 | 0.5 | 5 | 0.756 | 5373.00 |
| s1.1-7B | Adaptive Self Critique Sampler | 4096 | – | – | 0.792 | 4244.91 |
| **Max Thinking Tokens = 8192** | | | | | | |
| s1.1-7B | Greedy Policy | 8192 | – | – | 0.714 | 3917.57 |
| s1.1-7B | Accumulated Probability Policy | 8192 | 0.1 | – | 0.774 | 5077.64 |
| s1.1-7B | Accumulated Probability Policy | 8192 | 1 | – | 0.788 | 5405.76 |
| s1.1-7B | Accumulated Probability Policy | 8192 | 3 | – | 0.796 | 6631.65 |
| s1.1-7B | Last-Interval Budget Policy | 8192 | 0.01 | – | 0.794 | 11033.53 |
| s1.1-7B | Last-Interval Budget Policy | 8192 | 0.1 | – | 0.800 | 10712.60 |
| s1.1-7B | Last-Interval Budget Policy | 8192 | 0.5 | – | 0.784 | 10512.50 |
| s1.1-7B | N-Spike Counter Policy | 8192 | 0.01 | 1 | 0.798 | 5163.36 |
| s1.1-7B | N-Spike Counter Policy | 8192 | 0.01 | 3 | 0.800 | 5817.96 |
| s1.1-7B | N-Spike Counter Policy | 8192 | 0.01 | 5 | 0.802 | 6726.14 |
| s1.1-7B | N-Spike Counter Policy | 8192 | 0.1 | 1 | 0.798 | 5163.36 |
| s1.1-7B | N-Spike Counter Policy | 8192 | 0.1 | 3 | 0.806 | 5975.91 |
| s1.1-7B | N-Spike Counter Policy | 8192 | 0.1 | 5 | 0.804 | 8105.31 |
| s1.1-7B | N-Spike Counter Policy | 8192 | 0.5 | 1 | 0.798 | 5114.54 |
| s1.1-7B | N-Spike Counter Policy | 8192 | 0.5 | 3 | 0.804 | 7010.83 |
| s1.1-7B | N-Spike Counter Policy | 8192 | 0.5 | 5 | 0.810 | 9727.63 |
| s1.1-7B | Adaptive Self Critique Sampler | 8192 | – | – | 0.812 | 7330.56 |

Table 6: Performance of Qwen3 Models on the Maths500 Benchmark. This table compares the Baseline performance of Qwen3-8b and Qwen3-14b models against the 'Peak-Sampler-Critique' method under various configurations. The primary metrics are Maths500 accuracy and the average token count per problem. The results show that the Peak-Sampler-Critique method, particularly with k=3 critiques, a spike threshold of 0.25, and a larger context window, achieves the highest accuracy (0.950) while significantly reducing the token count compared to the baseline.

| Model | Method | Critiques (k) | Context Window | Spike Threshold | Maths500 Acc. | Avg. Tokens |
|---|---|---|---|---|---|---|
| | Baseline | N/A | 8k / 16k | N/A | 0.924 | 4683.00 |
| | Baseline | N/A | 16k / 32k | N/A | 0.934 | 5312.00 |
| | Peak-Sampler | 1 | 8k / 16k | 0.10 | 0.900 | 2166.10 |
| | Peak-Sampler | 3 | 8k / 16k | 0.10 | 0.908 | 2718.62 |
| | Peak-Sampler | 5 | 8k / 16k | 0.10 | 0.922 | 3108.73 |
| | Peak-Sampler | 1 | 8k / 16k | 0.25 | 0.914 | 2482.83 |
| **Qwen3-8b** | Peak-Sampler | 3 | 8k / 16k | 0.25 | 0.918 | 2882.83 |
| | Peak-Sampler | 5 | 8k / 16k | 0.25 | 0.922 | 3220.43 |
| | Peak-Sampler | 1 | 16k / 32k | 0.10 | 0.906 | 2227.50 |
| | Peak-Sampler | 3 | 16k / 32k | 0.10 | 0.916 | 2799.19 |
| | Peak-Sampler | 5 | 16k / 32k | 0.10 | 0.928 | 3370.71 |
| | Peak-Sampler | 1 | 16k / 32k | 0.25 | 0.914 | 2568.46 |
| | **Peak-Sampler** | **3** | **16k / 32k** | **0.25** | **0.944** | **3055.18** |
| | Peak-Sampler | 5 | 16k / 32k | 0.25 | 0.937 | 3635.18 |
| | Baseline | N/A | 8k / 16k | N/A | 0.933 | 4286.70 |
| | Baseline | N/A | 16k / 32k | N/A | 0.940 | 4732.62 |
| | Peak-Sampler | 1 | 8k / 16k | 0.25 | 0.928 | 2141.40 |
| **Qwen3-14b** | Peak-Sampler | 3 | 8k / 16k | 0.25 | 0.932 | 2683.92 |
| | Peak-Sampler | 5 | 8k / 16k | 0.25 | 0.930 | 2883.92 |
| | Peak-Sampler | 1 | 16k / 16k | 0.25 | 0.932 | 2138.05 |
| | **Peak-Sampler** | **3** | **16k / 32k** | **0.25** | **0.950** | **3084.97** |
| | **Peak-Sampler** | **5** | **16k / 32k** | **0.25** | **0.950** | **3524.97** |

Table 7: Performance of Qwen3 Models on the AIME25 Benchmark. This table presents the accuracy and average token consumption for Qwen3 models of varying sizes (4B, 8B, 14B). We compare the standard Baseline generation method against our 'Peak-Sampler-Critique' approach with an increasing number of critiques (k). The Peak-Sampler-Critique method generally improves accuracy over the baseline for all model sizes, with performance scaling with the number of critiques. The Qwen3-14B model with k=7 achieves the highest accuracy of 0.722, a notable improvement over its baseline performance of 0.655. All experiments were conducted using a 16k/32k context window and a spike threshold of 0.25 for the Peak-Sampler method.

| Model | Method | Critiques (k) | AIME Accuracy | Average Token Count |
|---|---|---|---|---|
| | Baseline | N/A | 0.578 | 13923.5 |
| | Peak-Sampler | 1 | 0.434 | 10047.20 |
| **Qwen3-4B** | Peak-Sampler | 3 | 0.500 | 12010.20 |
| | Peak-Sampler | 5 | 0.588 | 13451.10 |
| | **Peak-Sampler** | **7** | **0.600** | **14176.20** |
| | Baseline | N/A | $0.589 \pm 0.056$ | $13793.5 \pm 113.77$ |
| | Peak-Sampler | 1 | $0.500 \pm 0.081$ | $12364.1 \pm 320.96$ |
| **Qwen3-8B** | Peak-Sampler | 3 | $0.622 \pm 0.056$ | $13408.4 \pm 199.80$ |
| | **Peak-Sampler** | **5** | $\mathbf{0.667 \pm 0.027}$ | $\mathbf{13850.0 \pm 462.10}$ |
| | Peak-Sampler | 7 | $0.656 \pm 0.032$ | $15192.0 \pm 483.86$ |
| | Baseline | N/A | $0.655 \pm 0.041$ | $13278.0 \pm 201.50$ |
| | Peak-Sampler | 1 | $0.688 \pm 0.068$ | $11977.7 \pm 208.60$ |
| **Qwen3-14B** | Peak-Sampler | 3 | $0.667 \pm 0.072$ | $13099.4 \pm 142.80$ |
| | Peak-Sampler | 5 | $0.711 \pm 0.042$ | $13559.0 \pm 378.60$ |
| | **Peak-Sampler** | **7** | $\mathbf{0.722 \pm 0.031}$ | $\mathbf{14064.0 \pm 666.58}$ |

