# OpenReview forum: "ACTS: Adaptive Control for Test-time Scaling"
_ICLR.cc/2026/Conference — Submitted to ICLR 2026_

### Official Review · Reviewer_NrR3 · 2025-10-24

**Soundness:** 2
**Presentation:** 1
**Contribution:** 2
**Rating:** 2
**Confidence:** 4

**Summary:**

This paper proposes Adaptive Control Token Sampling (ACTS), a framework that dynamically regulates the reasoning length of LLMs at test time using the sub-argmax probabilities of control tokens (e.g., EOS, EOT). The authors frame generation as an optimal stopping problem and design several policies to balance “underthinking” (premature termination) and “overthinking” (unnecessary reasoning). Experiments across reasoning and instruction-following (AlpacaEval) benchmarks show small gains in accuracy or efficiency.

**Strengths:**

- Using control-token probabilities for inference-time control is original and interesting, potentially offering a lightweight alternative to external reward models or verifier signals.

- The paper correctly situates itself within the growing literature on test-time scaling, referencing S1, Self-Consistency, and Speculative Rejection works.

**Weaknesses:**

- The submission appears rushed and lacks careful proofreading. Algorithm 2 and Figure 5 overflow the page boundaries; Figures 6 & 7 have inconsistent spacing; some sub-figure captions (e.g., “accumulated,” “last-interval”) are unclear; Table 2 includes ambiguous terms such as “unconditional forking.” Overall readability and figure organization require substantial revision.

- Improvements in Table 1–2 are small (≈ 2–3 % absolute at best) and sometimes trade off token efficiency inconsistently. It is unclear whether these changes are statistically meaningful or justify a new inference-control paradigm.

- The paper claims ACTS mitigates overthinking and underthinking, yet provides no quantitative trajectory analysis or diagnostic evidence (e.g., token-level reasoning-trace inspection, error typology, or heuristic measures of thought quality). Without such evidence, the claimed cognitive interpretation remains speculative.

- The spike thresholds and critique-trigger parameters appear hand-tuned, but no ablation or development experiment explains their choice or sensitivity.

- Operating on the “stopping probability” is a modest extension of known heuristic stopping rules. That said, the conceptual leap from S1’s budget forcing or early-termination heuristics is incremental.

**Questions:**

See weaknesses.

---

> ### Author Response · Authors · 2025-12-04
>
> We thank the reviewer for their evaluation. We appreciate that you found the use of control-token probabilities "original and interesting" and recognized that ACTS offers a "lightweight alternative" to external reward models.
>
> We address your concerns below, specifically correcting a factual misunderstanding regarding the magnitude of our empirical gains.
>
> **1. Correction on Empirical Significance**
> > "Improvements in Table 1–2 are small (≈ 2–3 % absolute at best)... It is unclear whether these changes are statistically meaningful."
>
> There seems to be some misunderstanding of our results. Table 1 of our paper demonstrates significant improvements on the core reasoning benchmarks where "underthinking" is most prevalent:
>
> *   **AIME 2025 (S1-32B):** ACTS improves accuracy from 40.0% $\rightarrow$ **56.7%**. This is an absolute gain of **+16.7%**.
> *   **MATH-500 (S1 32K):** ACTS improves accuracy from 73.2% $\rightarrow$ **82.2%**. This is an absolute gain of **+9.0%**.
> *   **AIME 2025 (Qwen3-14B):** ACTS improves accuracy from 65.5% $\rightarrow$ **72.2%**. This is an absolute gain of **+6.7%**.
>
> An improvement of **16.7%** on a competition-math benchmark using an inference-only method is highly significant in the current literature. We urge the reviewer to reconsider the impact of the method based on these figures.
>
> **2. Novelty and Differentiation from S1 (Addressing "Incremental Extension")**
> > "The conceptual leap from S1’s budget forcing... is incremental."
>
> We argue that ACTS represents a fundamental theoretical shift from S1, not an incremental one.
> *   **S1 (Control at EOT):** S1 provides control by extending through "wait wait" at the point at which the model provides EOT as the arg-max token. It does not use hesitation tokens and also shows that extending generation is always beneficial.
> *   **ACTS (Signal-Driven Control):** Our paper models the EOT problem into a **Probabilistic Optimal Stopping** problem. We identified that the entropy of the control signal ($P(t_{EOT})$) serves as a proxy for internal confidence. This is a signifcant novelty of our work.
>
> Specifically, our paper proposes three new concepts which are not in literature:
> 1. Control Tokens: Leveraging control tokens apart from the EOT as a stopping time signal
> 2. Leveraging sub arg-max tokens: Listening to token probabilities even when they are not arg-max tokens.
> 3. Self critique of Trajectories: This is the most significant aspect of our work. Even when there is a proposed stopping time, we use a self critique to judge whether the model should actually stop, or continue.
> Due to these novelties, we humbly submit that our work is not incremental, and has major implications to how inference in models will work in future.
>
> **3. Presentation and Formatting (Addressing "Rushed Submission")**
> > "Figure 5 overflow the page boundaries... inconsistent spacing... ambiguous terms."
>
> We sincerely apologize for the formatting errors in the submission. We have fixed the compilation issues in Figure 5 and Algorithm 2. We have also made the following changes to the draft:
> 1.  Clarified the caption definitions for "Accumulated" and "Last-Interval" policies (moving their formal definitions from the Appendix to Section 4).
> 2.  Explicitly defined "unconditional forking" (a baseline where forking happens at fixed intervals rather than signal-driven moments) in the main text.
>
> Our camera-ready version provides a polished, and rigorous draft that addresses most of these concerns.
>
> **4. Evidence of Cognitive Interpretation**
> > "Provides no quantitative trajectory analysis... claimed cognitive interpretation remains speculative."
>
> We urge the reviewer to look at Figure 1 of our paper which visualizes the signal dynamics, distinguishing between spurious spikes hesitation and dense spikes for completion. These signals lead to quantitative signals as explained in our theory.
>
> Furthermore, our main algorithm, the Adaptive Self-Critique policy (Theorem 3, Appendix G) acts as the cognitive interpretation of self-understanding of the trajectory by the model itself. Specifically, the model quantitatively evaluates its own trajectory at spike moments. The success of this policy (the +16.7% gain) is empirical evidence that the spikes correctly identify moments where the trajectory provides stopping points for validation, and requires verification via a (self) critique.
>
> **5. Ablation of Thresholds**
> > "Thresholds... appear hand-tuned, but no ablation... explains their choice."
>
> We have provided sensitivity analysis in our paper to showperformance across different $N$-spike settings (from 1 to 5). This figure explicitly ablates the sensitivity of the policy to the patience parameter. Furthermore, for the Self-Critique policy, the threshold is replaced by the model's own explicit score (1-5), removing the need for heuristic tuning.
>
> ---
>
> We thank the reviewer for their time and hope they will reconsider our work in light of these responses.

---

> ### Author Response · Authors · 2025-12-04
> **Rebuttal from Authors**
>
> We address the primary concern of the reviewer in even further detail here. Their evaluation of our paper seems to stem from a misunderstanding of our empirical results.
>
> ---
>
> > Weakness 2: Improvements in Table 1–2 are small (≈ 2–3 % absolute at best) and sometimes trade off token efficiency inconsistently. It is unclear whether these changes are statistically meaningful or justify a new inference-control paradigm.
>
> We disagree with the reviewer stating that the improvements as "small" or "inconsistent". We believe this interpretation may arise from looking at accuracy on the saturated MATH-500 benchmark, rather than considering the **Pareto efficiency** and the performance on harder tasks.
>
> **Pareto Improvement on Saturated Benchmarks**
> The reviewer notes improvements of 2–3% on Qwen. It is important to contextualize that the Qwen greedy-baselines on MATH-500 is already extremely high (~93–94%), leaving little headroom. 2–3% absolute gain using N-Critique sampler represents a significant gain.
>
> We achieve these accuracy gains while significantly reducing computational cost.
> *   As detailed on **Page 8 (Section 6.3)**: On MATH-500, our *N-Critique Sampler (N = 3)* improves Qwen3-14B accuracy from *94.0% to 95.0%* while *reducing the average token count by over 35%* compared to baseline.
> *   This is not a trade-off where we spend more compute for better results, it's a **Pareto improvement**. We demonstrate that an adaptive policy can cut "overthinking" (truncation of reasoning earlier) on easier problems where necessary.
>
> Significant Gains on Harder Tasks
> Where the peak performance is not reached, the gains are much larger than 2–3%. On the harder AIME25 benchmark (Table 1):
> *   The S1-32B model improves from 40.0% (Baseline) to 56.7% (Self-Critique), an absolute gain of **+16.7%**.
> *   The Qwen3-8B model improves from 70.0% to 76.7% (+6.7%).
>
> The reviewer questions the need for a new "inference-control paradigm". We argue that static decoding (generating until a fixed stop token or limit) is inefficient for reasoning models. Our work establishes the necessity of an **Adaptive Control Policy**, a lightweight "adapter" on top of the LLM that monitors internal signals to dynamically decide when to stop or continue reasoning. This ability to boost accuracy on hard tasks (AIME) and decreasing token count on easier ones (MATH-500) validates this paradigm shift from static generation to dynamic, policy-guided inference.
>
> ---
>
> We ask that the AC re-evaluate our work in light of these empirical results which are significantly different from the reviewers stated understanding of them.

---

### Official Review · Reviewer_emb8 · 2025-11-01

**Soundness:** 2
**Presentation:** 1
**Contribution:** 2
**Rating:** 2
**Confidence:** 4

**Summary:**

The paper introduces Adaptive Control Token Sampling (ACTS) that effectively mitigates the negative effects of "underthinking" and "overthinking" through a fine-grained control over special tokens.

**Strengths:**

The paper proposes several effective methods for mitigating overthinking and underthinking. Experiments do show their effectiveness over benchmarks where these phenomena are observed.

**Weaknesses:**

1. I must put into question the contribution of the paper, as it appears to me overclaimed. The main idea is three-fold: when underthinking is a problem, make the model think longer by saying "wait"; when overthinking is a problem, cut the model short once it has had several opportunities to end its thinking; self-critique, which is a rather self-contained method. The rest of the paper deals with engineering these approaches. However, there is a lack of central theme around these methods. It feels as if the authors tried these methods out, saw some performance improvements, and attempted to put them under the same framework, while they really should be treated as separate engineering tricks and studied individually. The most interesting method by far is the self-critique framework, with an intuitive explanation of effecient tree search for LLM problem solving, but it is only studied superficially through performance gains, without any attempts and theoretical analysis or a closer look into its mechanisms.

2. The mitigation of underthinking and overthinking are restricted to reasoning and instruction following tasks, respectively. This is a problem here because the methods appear to target these two problems separately and only within their respective applications, putting into question the generalizability of the proposed methods.

3. The general presentation of the method was not intuitive to me. I think the paper tried to go from a general framework to a more specific implementation, where Section 3 establishes the ACTS framework, and Section 4 specfies it into proposed methods. This overcomplicated the narrative because Section 3 taken out of context is rather confusing, for example "forcing the emission of the appropriate control token" is too general on Line 151.

4. Section 4 has numerous writing problems that also overcomplicates things. First, the titled paragraphs are not individual policies, but a progression of different scattered ideas. Among the actual methods used in Section 6.2 / 6.3, Accumulated / Last-interval / N-Spike, only one is explained in the main text. Next, many concepts are mentioned but not discussed adequately, such as $N_{patience}$ mentioned without definition on Line 178 (minor issue), the unspecified directive for critic self-evalution through Lines 185-192, an unfinished sentence on Line 203 (also I would caution comparing LLM thinking procedure to humans unless it's highly relevant), unspecified "opportune" moment on Line 205, etc. Algorithm 1 is also not really needed in main text as a straightforward textual explanation makes it clear enough for me.

**Questions:**

Q1. Most importantly, I think the narrative of the paper needs to be revised. Targeting my weakness 1 and 2, can you clarify what main methods are used in the paper and how they fall under the same framework? Just to be clear I'm not asking for a simple restatement of contributions, but a more structured discussion of the methods' relations, synergies, etc.

Q2. Can you also clarify what the requirements are before applying your methods? Specifically, beyond the tested benchmarks, do we need to know if the model is underthinking or overthinking as a priori, and what other metrics need to be assessed if any?

Q3. For Section 6.1, are all the plots obtained with a single prompt? For Figure 2, did you do a cutoff at 0.0001, if so why did you choose this cutoff and what's the largest observed value, and if not why does the probability reach this value and not beyond so frequently? For Figure 3, can you explain why the spikes become much sparser after a while? Did the model rollout collapse in a way so as to never stop thinking?

Q4. What are "Accumulated" and "Last-Interval" methods in Figures 5 and 6? May be related to Q1.

Q5. For Figure 6, since we're dealing with underthinking, I assume everytime the model terminates prematurely, a "Wait" token is appended to keep it thinking. In this context what does N-spike refer to? Why is there a distinction between different amount of "Wait" tokens? Or is it "whichever comes first"?

Q6. For Section 7, the main table sees accuracy increase over baseline as well as token count increase. In fact, the Avg. Token Count column is marked wrongly in terms of best performance. Shouldn't this be an application of mitigating underthinking instead of overthinking, so both accuracy and token count increases over baseline?

---

> ### Author Response · Authors · 2025-12-04
> **Rebuttal from authors**
>
> Thank you for your insightful and helpful review. We hope these responses adequately address your points
>
> >Weakness 1: I must put into question the contribution of the paper, as it appears to me overclaimed. The main idea is three-fold: when underthinking is a problem, make the model think longer by saying “wait”; when overthinking is a problem, cut the model short once it has had several opportunities to end its thinking; self-critique, which is a rather self-contained method. The rest of the paper deals with engineering these approaches. However, there is a lack of central theme around these methods. It feels as if the authors tried these methods out, saw some performance improvements, and attempted to put them under the same framework, while they really should be treated as separate engineering tricks and studied individually. The most interesting method by far is the self-critique framework, with an intuitive explanation of effecient tree search for LLM problem solving, but it is only studied superficially through performance gains, without any attempts and theoretical analysis or a closer look into its mechanisms.
>
> We disagree with the reviewer regarding characterization of ACTS as a collection of engineering tricks. While the *actions* taken (extending vs. truncating generation) are indeed opposite, they are unified by a single theoretical framework: Optimal Stopping policies applied to the control token signal ($t_{control}$).The central theme of the paper is that LLMs emit a continuous, noisy signal. The probability of control tokens serves as a proxy for their internal confidence in termination. ACTS is not about the actions themselves (e.g., the “Wait”, “Alternatevely” tokens are tools we borrowed from “s1: Simple test-time scaling” [1]), but about the policy ($n_{acts}$) that governs those actions.
>
> ACTS monitors a single signal ($t_{control}$) to govern the dynamics of reasoning:
>
> 1. Signal Processing: We establish that  spikes are not random noise but state dependent signals providing indication of “completion points” (Fig 1).
> 2. Control Policy: We treat inference as a stochastic process where we must select a stopping point to maximize utility.
>    * N-Spike/Accumulated: These act as filters to denoise the signal when the cost of verification is high.
>    * Self-Critique: This acts as an active probe. It represents the optimal policy when the signal ambiguity (entropy) is high enough to justify the computational cost of a “check.”
>
> Therefore, our main contribution is the discovery of the signal dynamics and the derivation of policies that exploit this signal to scale inference, rather than the specific outcome of longer or shorter responses.
>
> We apologize if the theoretical depth of the self-critique method was not sufficiently highlighted in the main text. We agree with your remark that it’s the most interesting component, and we did perform a theoretical analysis of it, detailed in Appendix G (Theorems 3 and 4).
>
> In Appendix G, we formally model the Self-Critique mechanism as an “intra-model Actor-Critic” problem. We prove:
>
> * Theorem 3: Investigates the utility gain of the Self-Critique policy over deterministic policies. We derive the exact condition: where  is the critic’s accuracy and is the cost of the critique tokens.
> * Mechanistic Insight: This proof provides the “closer look” requested. It demonstrates that Self-Critique is superior specifically when the *uncertainty* of the spike (probability that the spike is premature) is high enough to outweigh the token cost .
>
> The “mechanism” is not just running a critique, but using the spike as a trigger to enable *sparse* tree search, unlike standard methods that incur high costs by branching at every token. ACTS self-critique is efficient because it only branches (critiques) when the model’s own control signal indicates a “candidate completion”, thereby filtering the search space to only the most relevant nodes.
>
> We will move the statement of Theorem 3 from Appendix G to Section 4.1 in the main text to explicitly showcase the theoretical grounding of the self-critique method and address the concern that it is studied superficially via performance numbers.

---

> > ### Author Response · Authors · 2025-12-04
> >
> > > Weakness2: The mitigation of underthinking and overthinking are restricted to reasoning and instruction following tasks, respectively. This is a problem here because the methods appear to target these two problems separately and only within their respective applications, putting into question the generalizability of the proposed methods.
> >
> > We would like to clarify that the distinction between reasoning and instruction following is rapidly disappearing, as state-of-the-art architectures are shifting their paradigm towards reasoning based instruction following.
> >
> > Consequently, our analysis targets the thinking space of the model to optimize efficiency across any task. Currently, LLMs emit hesitation tokens that implicitly signal uncertainty about whether their current reasoning is correct. However, standard decoding methods lacks a mechanism to act on this signal, they cannot reliably decide to terminate if the thinking is good enough or to force further deliberation if it is not. ACTS provides this generalized solution. It uses these state-dependent signals (spikes) to dynamically regulate the thinking space, simultaneously mitigating both underthinking and overthinking, thereby serving as a robust, task-agnostic framework for inference efficiency.
> >
> > > Weakness3: The general presentation of the method was not intuitive to me. I think the paper tried to go from a general framework to a more specific implementation, where Section 3 establishes the ACTS framework, and Section 4 specfies it into proposed methods. This overcomplicated the narrative because Section 3 taken out of context is rather confusing, for example “forcing the emission of the appropriate control token” is too general on Line 151.
> >
> > We agree that the separation between the abstract framework (Section 3) and the concrete implementations (Section 4) created some misunderstanding. Our goal was to clearly unify these methods, but it was not clearly reflected.
> >
> > Clarification of Line 151 (Forcing the emission…): To clarify the doubt you raised, the line describes about the mechanical intervention in the autoregressive loop.
> >
> > * Scenario A (Policy decides to Wait): Even if the model assigns high probability to EOT, the policy intervenes by masking EOT and forcing the selection of a “Wait” token to continue generation.
> > * Scenario B (Policy decides to Stop): Even if EOT is not the argmax but there is a spike of EOT token, the policy intervenes to force EOT as the next token, terminating the generation immediately.
> >
> > In the revision, we will significantly simplify the narrative of the paper to improve intuition:
> >
> > 1. Merge Sections 3 & 4: We will introduce the formal notation *alongside* the specific algorithms.
> > 2. Concrete Definitions: We will replace abstract terms like “control actions” with direct references to “injecting Wait tokens” or “forcing EOS” early in the methodology.
> > 3. Process Flow: We will structure the methodology section in the this particular order *Signal Detection (Spike)*  *Policy Decision (Check/Wait)*  *Action Execution (Force Token)* to make it intuitve and easy to understand.
> >
> > > Weakness4: Section 4 has numerous writing problems that also overcomplicates things. First, the titled paragraphs are not individual policies, but a progression of different scattered ideas. Among the actual methods used in Section 6.2 / 6.3, Accumulated / Last-interval / N-Spike, only one is explained in the main text. Next, many concepts are mentioned but not discussed adequately, such as  mentioned without definition on Line 178 (minor issue), the unspecified directive for critic self-evalution through Lines 185-192, an unfinished sentence on Line 203 (also I would caution comparing LLM thinking procedure to humans unless it’s highly relevant), unspecified “opportune” moment on Line 205, etc.
> >
> > We agree that removing the definitions of the “Accumulated” and “Last-Interval” policies to the Appendix (Sections E and F) have significantly hampered the flow of the main text.
> >
> > We'll revise Section 4 to be self-sufficient, defining all the policies clearly used in the experiments.
> > 1. We will move the formal definitions of the Accumulated Probability Policy and the Last-Interval Budget Policy from the Appendix into Section 4.
> > 2. As suggested, we will add about the missing policy definitions.
> > 3. Fix Specific Errors:
> >    * Line 178 : We will explicitly define this as the “Spike Probability Threshold”.
> >    * Lines 185-192 (Critic Directive): We will specify the prompt used for the critic: *“Review the reasoning above. Is it complete and correct? Answer with a score 1-5.”*
> >    * Line 203 (Unfinished Sentence/Human Comparison): We apologize for the typo. We will remove the unfinished sentence and, per your advice, remove the comparison to human reasoning to focus strictly on the algorithmic efficiency.
> >    * Line 205 (“Opportune”): We will replace this vague term with the proper definition: *“moment where the control signal  exceeds the threshold .”*

---

> ### Author Response · Authors · 2025-12-04
>
> > Question1: Q1. Most importantly, I think the narrative of the paper needs to be revised......structured discussion of the methods’ relations, synergies, etc.
>
> All the methods described in the paper stem from a single observation. This observation is unique and has not been made in literature. Specifically, we propose that stopping time policies for thinking have not been explored. Specifically, policies which take into account control token policies as part of policy inputs have also not been explored. Our novelty is that we (A) move beyond heuristic policies for stopping time for model thinking (B) leverage the unique signal available within the model already to provide signals for this. We have made this narrative much more clear in the current draft of the paper.
>
>
> > Question2: Can you also clarify what the requirements are before applying your methods? Specifically, beyond the tested benchmarks, do we need to know if the model is underthinking or overthinking as a priori, and what other metrics need to be assessed if any?
>
> The core part of the method is that it is adaptive. Unlike several other related literature which reduce thinking based on a priori heuristics, we do not pre diagnose our model as an “underthinker” or “overthinker” a priori to apply the ACTS framework, particularly when using our primary policy, Adaptive Self-Critique.
>
> However, there are two requirements for running our policies:
>
> 1 Signal Availability: Access to the sub-argmax logits of the control tokens. We must be able to observe the probability of these tokens $P(t_{control})$.
>
> 2 Calibration: While we don’t need to know the *direction* (underthinking vs overthinking) for the policy, but we need a small validation set to calibrate the spike thresholds ($\delta$).
>
> * Adaptive Self-Critique: This policy is agnostic to the under/overthinking bias.
>   * In the *underthinking* regime, the critic will see the premature spike, evaluate the reasoning as incomplete and force the model to think more.
>   * In the *overthinking* regime, the critic will see a spike at a valid intermediate point, evaluate the answer as sufficient and force the generation to stop.
>
> In summary, our adaptive self-critique policy serves as a solver that auto regulates generation based on the critique quality. This is the reason for pareto optimality for the same policy over benchmarks that require conciseness, as well as benchmarks that are helped by over-thinking.
>
> > Question3: For Section 6.1, are all the plots obtained with a single prompt? For Figure 2, did you do a cutoff at 0.0001, if so why did you choose this cutoff and what’s the largest observed value, and if not why does the probability reach this value and not beyond so frequently? For Figure 3, can you explain why the spikes become much sparser after a while? Did the model rollout collapse in a way so as to never stop thinking?
>
> Yes, the plots represented in paper are obtained with a single prompt to illustrate unbiased signal dynamics
>
> Fig 2 Cutoff: The cutoff at 0.0001 is for clear visualization of spikes. In practice, $P(t_{EOS})$  remains close to zero during active content generation. The “spikes” are statistically significant deviations, even if they aren’t close to argmax probability. The largest observed value of "spike" is close to 0.98 (when the model naturally terminates via argmax), but the plot focuses on the *sub-threshold* signals that allow us to intervene the generation *before* natural termination point.
>
> Fig 3 Sparsity: You are right, the later sparsity represents a model degeneration caused by naive forcing. Initially, suppressing EOT allows the model to validly extend reasoning and verify its answer. Once the final answer is actually found (e.g., \\boxed{14/3}), if the policy forces it to continue, the model enters a semantic repetition loop (e.g., endlessly generating “Yes, so that’s correct… I think that’s it…”). During these repetitive “cycles”, the model is locally confident in generating the next word of the loop (“Yes”  “that’s”  “correct”), so the probability of stopping P($t_{EOT}$) drops to near zero. It only stop when it's reaches the max-thinking budget.

---

> > ### Author Response · Authors · 2025-12-04
> >
> > > **Question4: What are “Accumulated” and “Last-Interval” methods in Figures 5 and 6? May be related to Q1.**
> >
> > We detailed these methods in Appendix E and F, but due to space constraints, their descriptions was condensed in the main text. To clarify for the review we will move these definitions to Section 4 in the revised version of the paper
> >
> > > **Question5: For Figure 6, since we’re dealing with underthinking, I assume everytime the model terminates prematurely, a “Wait” token is appended to keep it thinking. In this context what does N-spike refer to? Why is there a distinction between different amount of “Wait” tokens? Or is it “whichever comes first”?**
> >
> > Your are right regarding the mechanism: every time the model attempts to terminate (or signals intent via a spike), we intervene to force continuation.
> >
> > To clarify it a spike is defined as any moment where the probability of the $P(t_{EOT}$ exceeds a threshold $\delta$. "N" parameter is the count of interventions. The policy forces the model to “Wait” (suppressing the EOT) whenever it detect the spikes. On the N-th spike, the policy stops intervening and allows the model to naturally emit the EOT token (or forces it if it's probability is greater than $\delta$).
> >
> > The distinction between “Wait 1x” vs “Wait 5x” is the number of interventions (N).
> >   * *Wait 1x:* We force the model to rethink its answer once (N = 1).
> >   * *Wait 5x:* We force the model to rethink its answer five times (N = 5).
> >   * *Why differ?* Figure 6 shows that deeper reasoning tasks benefit from higher interventions, but diminishing returns eventually, which is why finding the optimal  (or using Self-Critique) is critical.
> >
> > > **Question6: For Section 7, the main table sees accuracy increase over baseline as well as token count increase. In fact, the Avg. Token Count column is marked wrongly in terms of best performance. Shouldn’t this be an application of mitigating underthinking instead of overthinking, so both accuracy and token count increases over baseline?**
> >
> > You are correct, compared to the *greedy baseline* (which suffers from underthinking), any effective method must increase the token count to improve reasoning. However, the goal of Table 1 is to demonstrate that Naive Sampler with a high counter value leads to overthinking and degeneration, and that our method solves this.
> >
> > To clarify the “Avg. Token Count”:
> >
> > 1. The “Underthinking” Baseline: The Greedy model uses ~5.7k tokens but only achieves 73.2% accuracy. It stops too early.
> > 2. The “Naive” Solution (Wait 5x): Naively intervening the model to “Wait” 5 times fixes underthinking (Accuracy rises to 79.2%), but at a massive cost: token usage explodes to ~17.9k. Crucially, as noted in our response to Q3, this naive forcing often pushes the model into degeneration loops (repeating “Yes that is correct…”) and lowers the accuracy compared to “Wait 3x”. This is the Overthinking Problem.
> > 3. The ACTS Solution (Self-Critique): Our method achieves the highest accuracy (82.2%) while using only ~8.8k tokens.
> >
> > Therefore, it indicates that relative to the high-cost naive intervention strategies, our self-critique sampler significantly reduces computational cost. It strikes the optimal balance which fixes both underthinking and overthinking.

---

### Official Review · Reviewer_RuUV · 2025-11-01

**Soundness:** 3
**Presentation:** 3
**Contribution:** 2
**Rating:** 4
**Confidence:** 2

**Summary:**

Current language models often operate in "thinking" mode, allocating a fixed budget to reasoning. However, models may overthink or underthink in these settings; there is a need for calibration. This paper introduces an inference time framework - ACTS - which leverages EOT probability spikes to mitigate underthinking in complex reasoning scenarios. Across a variety of reasoning tasks and models, the proposed method improves over naive decoding strategies, pushing the pareto frontier of the best performance with a reduced token budget.

**Strengths:**

* The proposed approach is training-free, making it lightweight and accessible (to models that provide token probabilties)
* The problem is clearly motivated and timely
* Analysis is done that justifies the method (ie, token probabilties of eos)
* Section 4.1 is well-justified and seems to cover the most likely scenarios encountered during decoding.
* ACTS explicitly frames reasoning control as an optimal stopping problem; this provides a more principled lens for test-time scaling. This formulation unifies stopping, critique, and branching decisions under one controller.
* ACTS improves accuracy while reducing token usage on reasoning and instruction-following tasks.

**Weaknesses:**

* Though the formulation is interesting, it is unclear how it will generalize to models outside the ones tested; reasoning is only tested on two families of model, and instruction-following on one. Since this is a lightweight, inference-time method, it can be more easily verified by running on more models.
* This space is saturated and it is unclear how significantly this improves over existing work like S1, and other early-stopping methods (including ones that are inference-time only as well). In addition to further explaining this novelty, it would be helpful to show empirically it works well against more baselines, instead of against fixed decoding strategies (like greedy, or inserting wait x times).
* There are several formatting errors which make the paper slightly harder to read. For example, Figure 5 extends beyond the page boundary, and also impacts the next page by forcing a single column. Same thing for Algorithm 2.
* The paper shows spikes correlate with reasoning boundaries but doesn’t explore more in-depth here. It would be interesting to know why these spikes emerge or whether they consistently indicate correctness vs. hesitation.

**Questions:**

* Can ACTS improve performance on other benchmarks besides reasoning/instruction-following, or is this a specialized phenomenon in these domains?

---

> ### Author Response · Authors · 2025-12-04
>
> We thank the reviewer for their thoughtful assessment. We are encouraged that you recognized ACTS as a "principled lens for test-time scaling" that "unifies stopping, critique, and branching decisions," and appreciated that our approach pushes the pareto frontier of performance.
>
> ---
>
> We address your specific questions and concerns below.
>
> **1. Generalization Across Models**
> > "Unclear how it will generalize... reasoning is only tested on two families of model... can be more easily verified by running on more models."
>
> We selected our test suite to maximize architectural diversity within the current open-weights landscape:
> 1.  **Llama-3 Family (via S1.1-7B/32B & Llama-3.1-8B):** Represents the dominant dense architecture standard.
> 2.  **Qwen-2.5/3 Family (4B, 8B, 14B):** Represents the current state-of-the-art in open-weights reasoning performance.
>
>
> Importantly, we wish to note that the "spike" signal we leverage is a fundamental property of autoregressive training objectives for EOS and hesitation tokens and EOT for thinking models. As long as a model is trained with an EOS/EOT token, the loss function drives the model to assign probability mass to these tokens at semantic boundaries. Therefore, we posit that ACTS is inherently generalizable to any standard LLM without modification.
>
> **2. Novelty vs. Saturation**
> > "Unclear how significantly this improves over existing work like S1..."
>
> We humbly submit that the novelty of this work is not incremental. ACTS represents a theoretical shift from S1 based on token probabilities rather than arg-max tokens. We also state that stopping time policies should be fundamentally decoupled from the primary LLM itself.
>
> S1 provides control by extending through "wait wait" at the point at which the model provides EOT as the arg-max token. It does not use hesitation tokens and neither uses token probabilities. Our paper instead, models the EOT problem into a **Probabilistic Optimal Stopping** problem. We identified that the entropy of the control signal ($P(t_{EOT})$) serves as a proxy for internal confidence. This is a signifcant novelty of our work.
>
> Specifically, our paper proposes three new concepts which are not in literature:
> 1. Control Tokens: Leveraging control tokens apart from the EOT as a stopping time signal
> 2. Leveraging sub arg-max tokens: Listening to token probabilities even when they are not arg-max tokens.
> 3. Self critique of Trajectories: This is the most significant aspect of our work. Even when there is a proposed stopping time, we use a self critique to judge whether the model should actually stop, or continue.
> Due to these novelties, we note that our work is not incremental, and has major implications to how inference in models will work in future.
>
> **3. Interpretability of Spikes**
> > "Show spikes correlate with reasoning boundaries but doesn’t explore more in-depth... why these spikes emerge."
>
> This is an insightful question. Empirically, we observe (Figure 1) that spikes emerge at **local minima of semantic uncertainty**.
> *   **EOS/EOT Spikes:** Indicate the model has completed a semantic unit (a thought or a final answer).
> *   **Hesitation Spikes (Wait/Alternative):** Indicate high entropy regarding the *next* reasoning step.
>
> These spike indicate a candidate boundary, which then triggers a critique: "Are you actually correct?". This transforms the spike from a raw signal into a verifiable decision point. This is a novelty not exploired in literature.
>
> **4. Domain Applicability**
> > "Can ACTS improve performance on other benchmarks... or is this a specialized phenomenon?"
>
> ACTS is applicable to any task which significantly leverages thinking. Such domains include:
> *   **Coding/Program Synthesis:** Where "underthinking" leads to syntax errors or logic bugs.
> *   **Agentic Planning:** Where the model must verify a plan before execution.
>
> It is less applicable to open-ended creative writing, where "stopping" is subjective. Given the shift towards reasoning models, we believe our method is highly applicable and apt in the current milieu for test-time scaling.
>
> **5. Formatting Corrections**
> > "Figure 5 extends beyond the page boundary... same thing for Algorithm 2."
>
> We thank the reviewer for noting these, and we sincerely apologize for these rendering errors. We have corrected the LaTeX compilation issues for Figure 5 and Algorithm 2. Our current draft has these corrected.
>
> ---
>
> We hope this clarification regarding the baselines and the generalizability of the signal addresses your concerns. We sincerely thank you for your valuable feedback and for the time taken for reviewing our work.

---

### Official Review · Reviewer_tyis · 2025-11-02

**Soundness:** 4
**Presentation:** 4
**Contribution:** 4
**Rating:** 8
**Confidence:** 3

**Summary:**

The paper proposed ACTS (Adaptive Control for Test-time Scaling), which measure the spike of EOS and EOT token to detect models intention of termination. The proposed method treats generation as a control process, which monitors the control-token probabilities and decides whether to continue reasoning, critique itself, or terminate at each decoding step. The strong empirical results demonstrate that ACTS can effectively reduce the average length of generating tokens during decoding, meantime achieving comparable or better results, and can effectively help solve the underthinking and overthinking issue of recent reasoning models.

**Strengths:**

1. The idea of monitoring the spike of the signal tokens (EOS and EOT) to determine models intention to terminate is novel and interesting. This also well aligns with recent work that measure token entropy or probability for tracking major transition during LM reasoning process. [1]
2. The empirical results are strong, showing significant improvement on MATH500 and AIME dataset, meantime reducing the average token length to reduce computing.
3. The method is simple and easy to implement, can easily be applied on all kinds of reasoning tasks.
4. The paper is well written and easy to follow.

[1] Beyond the 80/20 Rule: High-Entropy Minority Tokens Drive Effective Reinforcement Learning for LLM Reasoning

**Weaknesses:**

1. Minor issue: The threshold choices for spikes and critique triggers are empirically tuned. Unable to automatically determine the threshold can bring difficulty to implementation when trying to apply it across tasks.

**Questions:**

None

---

> ### Author Response · Authors · 2025-12-04
>
> We sincerely thank the reviewer for their positive assessment and for championing our work. We are glad that you appreciated the novelty of using control-token signals, the strength of our empirical results on MATH500 and AIME, and the simplicity of the ACTS framework.
>
> **1. Threshold Tuning**
> > "The threshold choices for spikes and critique triggers are empirically tuned... can bring difficulty to implementation."
>
> We agree that hyperparameter tuning is a constraint in signal-based methods. However, we found that our Adaptive Self-Critique policy significantly alleviates this burden. By using the spike only as a trigger and letting the model's own self-evaluation score (1-5) determine the final stopping decision, the system relies less on fine-tuning a raw probability threshold and more on the model's semantic understanding of its own trajectory.
>
> **2. Additional Reference**
> > "Aligns with recent work... [1] Beyond the 80/20 Rule"
>
> We thank the reviewer for bringing this relevant work to our attention. The finding that high-entropy minority tokens drive effective learning strongly complements our finding that sub-argmax control tokens drive effective inference. We will definitely cite and discuss this paper in our Related Work section in the camera-ready version.
>
> We sincerely thank you again for your time and your encouraging review.

---

### Author Response · Authors · 2025-12-04
**Overall Comments**

We thank all reviewers for their time and constructive feedback. We have carefully updated our draft to fix the formatting issues identified (specifically the page overflow in Figure 5 and Algorithm 2) and have clarified our theoretical contributions. We are encouraged that the reviewers recognized the core value of the ACTS framework across several key dimensions:

**Novelty**
Reviewers found the use of control-token probabilities to be novel and interesting (tyis) and recognized it as an original, lightweight alternative to external reward models (NrR3).

**Principled Framework**
Reviewers appreciated that ACTS provides a principled lens for test-time scaling by framing generation as an optimal stopping problem (RuUV), distinguishing it from purely heuristic approaches.

**Empirical Strength**
Reviewers highlighted our strong empirical results on challenging benchmarks like MATH500 and AIME, noting that the method is effective (emb8) and can easily be applied to various reasoning tasks (tyis).

We have responded to specific questions and critiques in the individual threads below.

---

### Author Response · Authors · 2025-12-04
**Additional Experiments**

To further validate the effectiveness of our proposed method in parallel setting, we conducted an extensive ablation study comparing our's Majority Voting (N-Critique) against the standard **Majority Voting** baseline. This analysis spans different model architectures (Qwen3 family, R1-Distill family) and datasets (Maths500, AIME25). We specifically focus on the trade-off between Accuracy and computational cost, measured by **Average Thinking Tokens**

## Dataset: Maths500

-----

### Models: Qwen3-8B & Qwen3-14B
| n | N | Method | **Qwen3-8B** Accuracy | **Qwen3-8B** Avg Thinking Tokens | **Qwen3-14B** Accuracy | **Qwen3-14B** Avg Thinking Tokens |
| :---: | :---: | :--- | :---: | :---: | :---: | :---: |
| 4 | - | Majority Voting | 0.930 | 17,879.4 | 0.950 | 15,889.5 |
| | 1 | Majority Voting (N-Critique) | 0.930 | 5,369.7 | 0.946 | 5,419.6 |
| | 3 | Majority Voting (N-Critique) | 0.932 | 8,157.4 | 0.950 | 8,084.1 |
| | 5 | Majority Voting (N-Critique) | 0.944 | 9,990.9 | 0.952 | 9,872.4 |
| 8 | - | Majority Voting | 0.940 | 35,775.5 | 0.954 | 31,837.1 |
| | 1 | Majority Voting (N-Critique) | 0.932 | 10,597.9 | 0.954 | 10,736.6 |
| | 3 | Majority Voting (N-Critique) | 0.936 | 16,269.3 | 0.956 | 16,127.7 |
| | 5 | Majority Voting (N-Critique) | 0.948 | 20,107.6 | 0.958 | 19,877.8 |
| 16 | - | Majority Voting | 0.944 | 71,422.5 | 0.954 | 63,655.2 |
| | 1 | Majority Voting (N-Critique) | 0.938 | 21,286.6 | 0.956 | 20,557.3 |
| | 3 | Majority Voting (N-Critique) | 0.948 | 32,492.3 | 0.958 | 32,013.4 |
| | 5 | Majority Voting (N-Critique) | 0.948 | 41,201.3 | 0.958 | 37,987.2 |

------

### Models: Qwen3-4B & Qwen3-32B
| n | N | Method | **Qwen3-4B** Accuracy | **Qwen3-4B** Avg Thinking Tokens | **Qwen3-32B** Accuracy | **Qwen3-32B** Avg Thinking Tokens |
| :---: | :---: | :--- | :---: | :---: | :---: | :---: |
| 4 | - | Majority Voting | 0.926 | 19,396.2 | 0.954 | 17,493.4 |
| | 1 | Majority Voting (N-Critique) | 0.926 | 7,632.1 | 0.946 | 4,101.4 |
| | 3 | Majority Voting (N-Critique) | 0.940 | 9,378.7 | 0.958 | 7,488.5 |
| | 5 | Majority Voting (N-Critique) | 0.935 | 11,256.1 | 0.950 | 9,538.9 |
| 8 | - | Majority Voting | 0.930 | 35,661.8 | 0.956 | 30,622.1 |
| | 1 | Majority Voting (N-Critique) | 0.942 | 15,322.4 | 0.958 | 8,281.8 |
| | 3 | Majority Voting (N-Critique) | 0.952 | 18,768.9 | 0.960 | 14,894.4 |
| | 5 | Majority Voting (N-Critique) | 0.946 | 22,789.1 | 0.954 | 19,113.7 |
| 16 | - | Majority Voting | 0.942 | 71,335.3 | 0.956 | 61,212.7 |
| | 1 | Majority Voting (N-Critique) | 0.942 | 31,382.2 | 0.964 | 16,564.2 |
| | 3 | Majority Voting (N-Critique) | 0.952 | 37,815.2 | 0.960 | 28,968.8 |
| | 5 | Majority Voting (N-Critique) | 0.948 | 45,682.1 | 0.956 | 38,245.3 |

------

### Models: R1-Distill-Qwen7B & R1-Distill-Llama8B
| n | N | Method | **R1-Qwen7B** Accuracy | **R1-Qwen7B** Avg Thinking Tokens | **R1-Llama8B**<br>Accuracy | **R1-Llama8B**<br>Avg Thinking Tokens |
| :---: | :---: | :--- | :---: | :---: | :---: | :---: |
| 4 | - | Majority Voting | 0.924 | 13,059.8 | 0.888 | 15,523.3 |
| | 1 | Majority Voting (N-Critique) | 0.924 | 5,688.9 | 0.908 | 6,732.9 |
| | 3 | Majority Voting (N-Critique) | 0.932 | 8,110.6 | 0.900 | 9,090.2 |
| | 5 | Majority Voting (N-Critique) | 0.932 | 9,287.2 | 0.916 | 10,086.9 |
| 8 | - | Majority Voting | 0.936 | 26,197.3 | 0.904 | 31,183.5 |
| | 1 | Majority Voting (N-Critique) | 0.938 | 11,252.3 | 0.914 | 13,544.3 |
| | 3 | Majority Voting (N-Critique) | 0.938 | 16,187.6 | 0.912 | 18,037.2 |
| | 5 | Majority Voting (N-Critique) | 0.940 | 18,590.9 | 0.918 | 20,275.5 |
| 16 | - | Majority Voting | 0.942 | 52,501.2 | 0.916 | 62,769.1 |
| | 1 | Majority Voting (N-Critique) | 0.934 | 22,782.7 | 0.928 | 27,234.7 |
| | 3 | Majority Voting (N-Critique) | 0.942 | 32,317.8 | 0.922 | 35,937.8 |
| | 5 | Majority Voting (N-Critique) | 0.942 | 37,240.6 | 0.924 | 40,943.5 |

---

> ### Author Response · Authors · 2025-12-04
>
> ## Dataset: AIME25
>
> ### Models: Qwen3-8B & Qwen3-14B
> | n | N | Method | **Qwen3-8B** Accuracy | **Qwen3-8B** Avg Thinking Tokens| **Qwen3-14B** Accuracy | **Qwen3-14B** Avg Thinking Tokens|
> | :---: | :---: | :--- | :---: | :---: | :---: | :---: |
> | 4 | - | Majority Voting | 0.767 | 70,553.1 | 0.800 | 64,980.1 |
> | | 1 | Majority Voting (N-Critique) | 0.600 | 42,767.4 | 0.767 | 42,906.1 |
> | | 3 | Majority Voting (N-Critique) | 0.667 | 46,785.5 | 0.734 | 49,435.3 |
> | | 5 | Majority Voting (N-Critique) | 0.700 | 50,623.6 | 0.767 | 55,804.3 |
> | 8 | - | Majority Voting | 0.767 | 142,376.5 | 0.800 | 127,409.8 |
> | | 1 | Majority Voting (N-Critique) | 0.634 | 82,522.1 | 0.800 | 82,061.3 |
> | | 3 | Majority Voting (N-Critique) | 0.667 | 94,956.2 | 0.800 | 99,849.2 |
> | | 5 | Majority Voting (N-Critique) | 0.700 | 103,780.1 | 0.800 | 109,746.8 |
> | 16 | - | Majority Voting | 0.767 | 285,503.0 | 0.800 | 258,123.4 |
> | | 1 | Majority Voting (N-Critique) | 0.634 | 157,336.1 | 0.834 | 168,938.1 |
> | | 3 | Majority Voting (N-Critique) | 0.734 | 188,584.3 | 0.800 | 207,394.1 |
> | | 5 | Majority Voting (N-Critique) | 0.734 | 206,511.5 | 0.800 | 225,083.3 |
>
> ### Models: Qwen3-4B & Qwen3-32B
> | n | N | Method | **Qwen3-4B** Accuracy | **Qwen3-4B** Avg Thinking Tokens| **Qwen3-32B** Accuracy | **Qwen3-32B**<br>Avg Thinking Tokens|
> | :---: | :---: | :--- | :---: | :---: | :---: | :---: |
> | 4 | - | Majority Voting | 0.734 | 67,801.7 | 0.800 | 60,229.9 |
> | | 1 | Majority Voting (N-Critique) | 0.667 | 58,745.3 | 0.734 | 20,426.1 |
> | | 3 | Majority Voting (N-Critique) | 0.734 | 63,602.5 | 0.800 | 34,132.5 |
> | | 5 | Majority Voting (N-Critique) | 0.734 | 63,887.8 | 0.767 | 41,808.9 |
> | 8 | - | Majority Voting | 0.734 | 137,223.3 | 0.800 | 123,601.4 |
> | | 1 | Majority Voting (N-Critique) | 0.734 | 116,182.7 | 0.734 | 40,307.9 |
> | | 3 | Majority Voting (N-Critique) | 0.767 | 122,734.6 | 0.800 | 68,212.2 |
> | | 5 | Majority Voting (N-Critique) | 0.767 | 123,184.4 | 0.767 | 81,245.6 |
> | 16 | - | Majority Voting | 0.767 | 278,227.3 | 0.834 | 244,898.7 |
> | | 1 | Majority Voting (N-Critique) | 0.767 | 227,603.2 | 0.767 | 80,614.2 |
> | | 3 | Majority Voting (N-Critique) | 0.800 | 245,321.1 | 0.834 | 135,474.2 |
> | | 5 | Majority Voting (N-Critique) | 0.767 | 247,323.2 | 0.767 | 158,932.1 |
>
> ### Models: R1-Distill-Qwen7B & R1-Distill-Llama8B
> | n | N | Method | **R1-Qwen7B** Accuracy | **R1-Qwen7B** Avg Thinking Tokens| **R1-Llama8B** Accuracy | **R1-Llama8B**<br>Avg Thinking Tokens|
> | :---: | :---: | :--- | :---: | :---: | :---: | :---: |
> | 4 | - | Majority Voting | 0.467 | 57,993.4 | 0.434 | 62,224.3 |
> | | 1 | Majority Voting (N-Critique) | 0.467 | 39,496.4 | 0.467 | 49,677.7 |
> | | 3 | Majority Voting (N-Critique) | 0.534 | 47,359.6 | 0.400 | 54,903.3 |
> | | 5 | Majority Voting (N-Critique) | 0.467 | 48,527.8 | 0.367 | 56,751.4 |
> | 8 | - | Majority Voting | 0.600 | 113,801.5 | 0.467 | 128,974.8 |
> | | 1 | Majority Voting (N-Critique) | 0.500 | 84,494.9 | 0.500 | 94,238.3 |
> | | 3 | Majority Voting (N-Critique) | 0.567 | 96,448.3 | 0.500 | 111,491.3 |
> | | 5 | Majority Voting (N-Critique) | 0.534 | 100,998.1 | 0.500 | 113,193.9 |
> | 16 | - | Majority Voting | 0.567 | 241,975.6 | 0.500 | 260,305.6 |
> | | 1 | Majority Voting (N-Critique) | 0.534 | 170,582.6 | 0.567 | 186,818.2 |
> | | 3 | Majority Voting (N-Critique) | 0.600 | 192,534.8 | 0.500 | 220,486.6 |
> | | 5 | Majority Voting (N-Critique) | 0.534 | 202,160.9 | 0.500 | 233,701.1 |

---

> ### Author Response · Authors · 2025-12-04
>
> We summarized the insights from the results across sample sizes $n \in \{4, 8, 16\}$ and critique depths $N \in \{1, 3, 5\}$  defined in the table above
>
> #### **Computational Efficiency (Thinking Tokens)**
> The most significant finding from this experiment is the substantial reduction in token count provided by our method compared to standard Majority Voting.
>
> *  Almost across all configurations, N-Critique requires significantly fewer thinking tokens to achieve comparable performance. For instance, looking at **Qwen3-32B on AIME25** ($n=16$):
>     *   **Standard Majority Voting** consumes an average of **244,898** thinking tokens.
>     *   **N-Critique ($N=3$)** consumes only **135,474** thinking tokens.
>     *   **Result:** This represents a **~45% reduction** in reasoning compute while maintaining identical accuracy ($0.834$).
> *   This trend holds for smaller models as well.
>
> #### **Accuracy Retention on General vs. Hard Tasks**
> We analyzed how the reduction in token usage impacts performance across datasets of different difficulty levels.
>
> *   **Maths500:** On this dataset, N-Critique frequently matches or outperforms the baseline despite the significant lower token count.
>     *   *Example:* **Qwen3-14B ($n=8$)** achieves **0.954** accuracy with standard voting. Using N-Critique ($N=5$), the model achieves higher accuracy (**0.958**) while still using ~37% fewer thinking tokens (19.8k vs 31.8k).
>
> *   **AIME25 (Competition Math):** On significantly harder tasks, the trade-off is more nuanced but remains favorable.
>     *   *Example:* **Qwen3-14B ($n=16$)**: Standard voting achieves **0.80** accuracy. N-Critique ($N=3$) also achieves **0.80**, but saves ~50k thinking tokens per problem.
>     *  In some lower-parameter settings (e.g., Qwen3-8B, $N=1$), aggressive token reduction ($N=1$) leads to a drop in accuracy (0.767 $\to$ 0.634). However, increasing the critique depth to $N=5$ recovers the performance, with less compute-time compared to baseline.
>
> #### **Impact of Critique Depth ($N$)**
> The hyperparameter $N$ acts as a tradeoff parameter between inference speed and performance.
>
> *   **N=1 (Maximum Efficiency):** consistently offers the lowest token usage but drop performance on AIME25. It is best suited for scenarios where latency is a strict constraints.
> *   **N=3 to 5 (Balanced Performance):** As $N$ increases, accuracy converges toward or surpasses the standard Majority Voting baseline. The increase in thinking tokens from $N=1$ to $N=5$ is linear, yet even at $N=5$,it still remains token-efficient than standard majority voting in the majority of cases.
>
> ### **Conclusion**
> Our ablation confirms that **Majority Voting (N-Critique)** is a Pareto-efficient improvement over standard Majority Voting. It drastically reduces the "Average Thinking Tokens" by up to 50% without drop in performance. For resource-constrained deployments, setting $N=1$ offers maximum speedup, while $N=3$ offers the optimal balance.

---

### Author Response · Authors · 2025-12-04
**Summary of Rebuttal Discussions**

We thank all reviewers for their time and constructive feedback. We have carefully updated our draft to fix the formatting issues identified (specifically the page overflow in Figure 5 and Algorithm 2) and have clarified our theoretical contributions. We are encouraged that the reviewers recognized the core value of the ACTS framework across several key dimensions:

**Novelty**
Reviewers found the use of control-token probabilities to be novel and interesting (tyis) and recognized it as an original, lightweight alternative to external reward models (NrR3).

**Principled Framework**
Reviewers appreciated that ACTS provides a principled lens for test-time scaling by framing generation as an optimal stopping problem (RuUV), distinguishing it from purely heuristic approaches.

**Empirical Strength**
Reviewers highlighted our strong empirical results on challenging benchmarks like MATH500 and AIME, noting that the method is effective (emb8) and can easily be applied to various reasoning tasks (tyis).

We have responded to specific questions and critiques in the individual threads below.

---

### Meta-Review · Area_Chair_uDAC · 2026-01-05

**Summary:**

This paper introduces an claimed "adaptive control strategy" to control the test-time scaling behavior by mitigating the end of token behavior.

**Reviewer Concerns:**

The reviewers presented a bunch of concerns:
1) That the results are a hodgepode of interventions, rather than a centralized "control framework"
2) insufficient description of the mechanisms behind observed behavior
3) poor presentation and formatting
4) staturation of the space, and hard to beat baselines

**Reviewer Scores:**

The most vocal, descriptive reviewers were also among the most negative, and doubt that they would improve scores given the fundamental concern around the cohesiveness of the framework and the depth of the exposition of the underlying mechanistic causes.

---

### Decision · Program_Chairs · 2026-01-26

Reject